



# Sources and formation mechanisms of carbonaceous aerosol at a regional background site in the Netherlands: Insights from a year-long radiocarbon study

Ulrike Dusek[1,2], Regina Hitzenberger[3], Anne Kasper-Giebl[4], Magdalena Kistler[4], Harro A. J. Meijer[2], Sönke Szidat[5], Lukas Wacker[6], Rupert Holzinger[1], Thomas Röckmann[1]

[1] Institute for Marine and Atmospheric research Utrecht (IMAU), Utrecht University, The Netherlands

[2] Centre for Isotope Research (CIO), University of Groningen, Groningen, The Netherlands

[3] Faculty of Physics, University of Vienna, Vienna, Austria

[4] Environmental and process analytics, Vienna University of Technology, Vienna, Austria

[5] Laboratory for Radiochemistry and Environmental Chemistry, University of Bern, Switzerland

[6] Laboratory of Ion Beam Physics, ETH Zürich, Switzerland

*Correspondence to*: Ulrike Dusek (u.dusek@rug.nl)

Keywords: carbonaceous aerosol, radiocarbon, source apportionment, PM2.5

**Abstract.** We measured the radioactive carbon isotope $^{14}C$ (radiocarbon) in various fractions of the carbonaceous aerosol sampled between February 2011 and March 2012 at the Cesar observatory in the

Netherlands. Based on the radiocarbon content in total carbon (TC), organic carbon (OC), water insoluble organic carbon (WIOC), and elemental carbon (EC), we estimated the contribution of major sources to the carbonaceous aerosol. The main source categories were fossil fuel combustion, biomass burning and other contemporary carbon, which is mainly biogenic secondary organic aerosol material (SOA).

A clear seasonal variation is seen in EC from biomass burning ($EC_{BB}$), with lowest values in summer and highest values in winter, but $EC_{BB}$ is a minor fraction of EC in all seasons. WIOC from contemporary sources is highly correlated with $EC_{BB}$, indicating that biomass burning is the dominant source of contemporary WIOC. This suggests that most biogenic SOA is water-soluble and that water insoluble carbon stems mainly from primary sources. Seasonal variations in other carbon fractions are

less clear and hardly distinguishable from variations related to air mass history.

Air masses originating from the ocean sector presumably contain little carbonaceous aerosol from outside the Netherlands, and during these conditions measured carbon concentrations reflect regional sources. In these situations absolute TC concentrations are usually rather low, around 1.5 µg m$^{-3}$ and $EC_{BB}$ is always very low ($\sim 0.05$ µg m$^{-3}$), even in winter, indicating that biomass burning is not a strong

source of carbonaceous aerosol in the Netherlands. In continental air masses, which usually arrive from the East or South and have spent several days over land, TC concentrations are on average by a factor





of 3 higher. $EC_{BB}$ increases more strongly than TC to 0.2 µg m$^{-3}$. Fossil EC and fossil WIOC, which are indicative of primary emissions, show a more moderate increase by a factor of 2.5 on average. An interesting case is fossil water soluble organic carbon (WSOC, calculated as OC-WIOC), which can be regarded as a proxy for SOA from fossil precursors. Fossil WSOC has low concentrations when

regional sources are sampled and increases by more than a factor of 5 in continental air masses. A longer residence time of air masses over land seems to result in increased SOA concentrations from fossil origin.

## 1 Introduction

Carbonaceous material constitutes a significant fraction of the atmospheric aerosol in almost all environments (e.g., Fuzzi et al., 2015; Pöschl, 2005; Putaud et al., 2004). With the continuing reduction of inorganic aerosol constituents in Europe (e.g., Chin et al., 2014), it will likely be the dominant aerosol component in the future. For further improvements in air quality, it is necessary to target the carbonaceous aerosol fraction. However, source apportionment of the organic aerosol fraction is

notoriously difficult, due to the large number of constituents, and the complexity of chemical formation and transformation processes of the organic aerosol (Fuzzi et al., 2006).

Analysis of the radioactive carbon isotope $^{14}$C in various carbonaceous aerosol fractions has become an important tool for source apportionment in the recent years (e.g., Currie, 2000; Gelencsér et al., 2007; Heal et al., 2011a and references therein; Szidat et al., 2006). The success of this method lies in its

clear-cut separation between fossil and contemporary sources of carbonaceous aerosol. $^{14}$C is continually produced in the upper troposphere and stratosphere. Thermal neutrons produced by cosmic rays react with $^{14}$N to $^{14}$C, which is oxidized via $^{14}$CO to $^{14}$CO$_2$. $^{14}$C decays with a half-life of 5,730 years (Godwin, 1962). In the living biosphere it is continually replenished from atmospheric $^{14}$CO$_2$, so that a typical contemporary level is established. Aerosol carbon originating from the living biosphere,

such as plant emissions or wood combustion, has therefore the contemporary $^{14}$C signature. Fossil fuels, however, have been buried for so long that $^{14}$C has completely decayed and as a consequence aerosol carbon from fossil fuels contains no $^{14}$C.

In aerosol source apportionment, the $^{14}$C content of a sample is usually reported as Fraction Modern (Mook and van der Plicht, 1999; Reimer et al., 2004):

$$F^{14}C = \frac{\left(\frac{^{14}C}{^{12}C}\right)_{sample}}{\left(\frac{^{14}C}{^{12}C}\right)_{1950}}, \qquad (1)$$

which relates the $^{14}$C/$^{12}$C ratio of the sample to the ratio of the unperturbed atmosphere in the year 1950. Both ratios are normalized to a d$^{13}$C value of -25 ‰. Aerosol carbon derived from living biomass should therefore have $F^{14}C \sim 1$, in an atmosphere unperturbed by human activities, whereas carbon from fossil sources has $F^{14}C = 0$. Human activities however, especially the atomic bomb tests, which





nearly doubled the natural $^{14}CO_2$ levels in the 1960s and 1970s, are perturbing the natural equilibrium. Currently, the atmospheric $CO_2$ has $F^{14}C$ of approximately 1.04 (Levin et al., 2010) which is decreasing every year, because the $^{14}CO_2$ produced by bomb testing is taken up by oceans and the biosphere. Moreover, fossil fuel consumption introduces $^{14}C$ free $CO_2$ into the atmosphere, leading to a

further decrease in atmospheric $F^{14}C$. This has consequences for the $F^{14}C$ of contemporary aerosol sources: Biogenic primary and secondary organic aerosols, as well as aerosols from cooking emissions have $F^{14}C$ close to the value of current atmospheric $CO_2$. $F^{14}C$ of aerosol from wood combustion is higher than that, because a significant fraction of carbon in the wood burned today was fixed during times when $^{14}C$ levels in the atmosphere were high. Estimates based on tree growth models (e.g., Lewis

et al., 2004; Mohn et al., 2008) range from 1.08 to 1.30 for biomass combusted in Western Europe (Genberg et al., 2011; Gilardoni et al., 2011; El Haddad et al., 2011; Minguillón et al., 2011; Szidat et al., 2006, 2007, 2009).

If $F^{14}C$ is separately measured on different carbon sub-fractions, such as organic carbon (OC) and elemental carbon (EC), then the three major sources of carbonaceous aerosol can be separated, namely

biogenic, biomass combustion, and fossil fuel combustion (Bernardoni et al., 2013; Minguillón et al., 2011; Szidat et al., 2006). For example, Szidat et al., (2007) showed that residential biomass combustion is the dominant source of carbonaceous aerosol in Alpine valleys in winter. Gustafsson et al., (2009) could demonstrate that in the brown haze over South Asia, the majority of the carbonaceous aerosol originates from biomass burning sources.

Measurements of $F^{14}C$ in either water insoluble OC (e.g., Szidat et al., 2006), or water soluble OC (Kirillova et al., 2010) can help to distinguish secondary from primary fossil OC. Primary OC/EC ratios for fossil sources are relatively well characterized and tend to produce mainly WIOC. Therefore fossil WIOC can be taken as indicative of primary fossil OC material and the fossil WSOC as indicative of fossil SOA material. Using this modified tracer approach Zhang et al., (2014) showed that

secondary fossil OC clearly exceeds primary fossil OC in Southern China. Measurements downwind of urban centers in California (Zotter et al., 2014a) and Japan (Morino et al., 2010) have highlighted the rapid formation of fossil SOA, using $^{14}C$ measurements together with positive matrix factorization of Aerosol Mass Spectrometer data and a chemical mass balance model, respectively.

The radiocarbon content of different sub-fractions of the carbonaceous aerosol has therefore the

potential to give information not only on sources but also formation pathways and processes of the organic aerosol. In this study we apply radiocarbon measurements to study the sources and formation processes of organic aerosol at a regional background site in the Netherlands, a heavily industrialized region with a high population density and generally high aerosol concentrations, where PM 10 values of larger than 50 $\mu g\ m^{-3}$ were still observed 20 – 30 times in the year 2011 on many measurements

stations. Carbonaceous material contributes on average around 30% to the total PM2.5 at regional background locations with somewhat higher concentrations in urban centers and at curbside locations (Weijers et al., 2011). Only few radiocarbon measurements have been reported for the aerosol in the Netherlands. A fraction modern of organic carbon ($F^{14}C_{(OC)}$) of around 0.7 has been measured at urban


background sites both in Amsterdam (Dusek et al., 2013) and in Rotterdam (Keuken et al., 2013) in spring, which is in the typical range of western European cities (Heal et al., 2011; Szidat et al., 2006, 2009). This is somewhat surprising for a heavily industrialized region like the Netherlands, where there is little residential wood combustion and few forests that emit large amounts of biogenic SOA

precursors. $F^{14}C_{OC}$ of total suspended particles at a coastal site in the Netherlands was considerably higher (0.86 on average) (Dusek et al., 2013).

These previous studies were limited in duration and only representative for a small geographical region. For the current study, we chose a regional background site surrounded by the biggest urban centers of the Netherlands and collected aerosol samples for more than one year. Due to the location of

the site and the sampling approach, which accounted for transport of air masses, the results presented in this study reflect seasonal changes as well as differences between regional air masses and pollution due to long-range transport.

**2 Methods**

**2.1 Measurement site**

Measurements were taken at the Cesar Observatory (http://www.cesar-observatory.nl/), which is located in the Western part of the Netherlands in an agricultural region. The immediate surroundings are dominated by open pastures with little variation in surface elevation and few wind breaks. 20–30 km to the North West and South East lie the cities of Utrecht and Rotterdam, respectively, and at a distance of around 50km to the West and North the cities of Den Haag, Leiden, and Amsterdam. The

closest highways connecting these major cities are at a distance of 10–20 kilometers from the site. This site therefore represents the regional background of a relatively polluted area in Western Europe.

**2.2 Sampling and filter handling**

Samples of particulate matter with a diameter less than 2.5 μm were taken with a high volume sampler (Digitel DHA-80) from February, 2011 to March 2012. The sampling duration varied between 2 and 7

days. Since the concentration and properties of organic aerosol can be significantly influenced by long-range transport, the start and end dates of each sampling were chosen based on air mass back-trajectory forecasts calculated every day by the NOAA-Hysplit model (Stein et al., 2015). This way, individual filter samples generally contain aerosol from specific source regions.

Aerosol particles were collected on circular quartz fiber filters (Whatman QMA1851-150) with a

diameter of 15 cm. The filters were pre-heated at 800 °C overnight to remove adsorbed organic contaminants. Before and after sampling the filters were kept wrapped in pre-backed aluminum foil and stored at -20°C, except during transport to and from the sampling site (approx. 1hr). After sampling, loaded filters were immediately removed from the filters stacks to avoid adsorption of volatile organic compounds to the quartz fibers. All instruments that came in contact with the filters (e.g. cutters,





tweezers) were pre-cleaned with acetone followed by ethanol. Blank filters were treated exactly like the sample filters, except that they were kept in the sampler only for a few minutes without switching it on.

### 2.3 Combustion of carbon fractions and standards

The separation of OC and EC for all filter samples was performed on the aerosol combustion system (ACS) that was developed at the University of Utrecht and has been described in detail by Dusek et al., (2014). Briefly, the ACS consists of a combustion tube, where aerosol samples are combusted at different temperatures in pure $O_2$, and a purification line where the resulting $CO_2$ is isolated and separated from other gases, such as water vapor or $NO_x$. The purified $CO_2$ is then stored in flame-sealed glass ampoules until [14]C analysis.

We analyze radiocarbon in the following carbon fractions: Total carbon (TC), OC, WIOC and EC. For TC combustion an aliquot of the filter is heated at 650 °C for 15 minutes. The carbon fraction considered representative of OC is combusted by heating of a different filter piece at 360 °C for 15 min. For WIOC a water-extracted (see: Dusek et al., 2014) filter piece is heated at 360 °C for 15 min.

EC is combusted, after OC has been completely burned off the filter. To achieve completely OC removal, WSOC is first removed from the filter by water extraction to prevent charring of organic material (Dusek et al., 2014). Subsequently, most WIOC is removed by heating the filter piece at 360 °C for 15 min. Then the oven temperature is raised to 450 °C for two minutes and in this step a mixture of the most refractory OC and EC is burned off the filter. The remaining EC is then combusted by

heating at 650 °C. Zhang et al., (2012) and Dusek et al., (2014) estimate that after water extraction and flash combustion in $O_2$, charred organic compounds contribute at most 5% to the recovered EC. A mean bias for charring of 0.04 is therefore subtracted from the fraction modern of EC ($F^{14}C_{(EC)}$) before source apportionment.

As quality control two sets of standards with known [14]C content are analyzed at regular intervals: the HOxII oxalic acid standard with a nominal $F^{14}C$ of 1.3408 and a graphite standard with $F^{14}C$ of 0. The standards are directly put on the filter holder and heated at 650 °C for 15 minutes. From the deviation of the measured $F^{14}C$ from the nominal values, the contamination introduced by the combustion procedure can be estimated. This contamination is on average below 3 μg C / combustion (Dusek et al.,

2014) and relatively small compared to typical sample amounts between 50 and 200 μg C.

### 2.4 [14]C measurement and data correction

After purification the $CO_2$ collected from the combustion of various aerosol fractions on the ACS system is sealed in glass ampoules. Most of these $CO_2$ samples were sent to the Centre for Isotope Research (CIO) at the University of Groningen for graphitization and AMS measurements. The $CO_2$

extracted from all samples collected in spring 2011 and several blanks and standards were analyzed without graphitization at the AMS facility at the Laboratory of Ion Beam Physics of ETH Zürich.



At CIO $CO_2$ is reduced to graphite by reaction with molecular hydrogen at a molecular ratio $H_2/CO_2$ of 2.5. A porous iron pellet (de Rooij et al., 2010) at a temperature of 600°C is used as a catalyst for the reaction. The resulting water vapor is cryogenically removed using Peltier cooling elements. The yield of graphite is virtually 100% for samples larger than 30 µg. After graphite had formed on the iron

pellet, it is pressed into a 1.5 mm target holder, which is introduced into the AMS system for subsequent measurement. The AMS system (van der Plicht et al., 2000) is dedicated to $^{14}$C analysis, and simultaneously measures $^{13}$C/$^{12}$C and $^{14}$C/$^{12}$C ratios. Samples below 500 mg are analyzed together with varying amounts of reference materials ranging from 50 – 500 µg C. Two reference materials with known $^{14}$C content are used: the HOxII standard ($F^{14}C = 1.3407$) and a $^{14}$C-free $CO_2$ gas ($F^{14}C = 0$).

The differences between actually measured and nominal $F^{14}C$ values are used for correcting the sample values (de Rooij et al., 2010) for contamination during graphitization and AMS measurement. The contamination is typically below 2 µg C (Prokopiou, 2010).

At ETH Zürich, $^{14}$C is measured directly without graphitization from the purified $CO_2$ using the AMS system MICADAS with a gas ion source (Ruff et al., 2007). In an automated gas interface, the $CO_2$ is

released from ampoules with a cracker, mixed with helium and transferred into the ion source with a constant gas flow (Wacker et al., 2013). This system allows $^{14}$C analysis of ultra-small samples larger than 2 µg C. Unknown samples are corrected for blank and isotope fractionation as well as normalized to $F^{14}C$ values using HOxII and a $^{14}$C-free material as gaseous standards. A constant contamination of <0.1 µg C with an $F^{14}C$ value of 0.5 was determined for the $^{14}$C gas analysis (Ruff et al., 2010), which

was applied for the correction of the unknown samples.

**2.5 Blank correction**

The amount of carbon on a blank filter was on average 0.34 µg cm$^{-2}$ for OC, based on the analysis of three blank filters. The amount of OC on sample filters varied from roughly 10 to 100 mg cm$^{-2}$, with an average of around 30 µg cm$^{-2}$. Due to the small amount of carbon on the blank filters it was only

possible to analyze $^{14}$C for one of these blank filters. The $F^{14}C_{(OC)}$ on this blank filter was 0.797 ± 0.019. The amount of blank EC is mainly added during the combustion process, and is therefore not necessarily proportional to the area of the filter used for analysis. Blank EC was found to be 0.3 ± 0.1 µg cm$^{-2}$ for a filter piece of 7 cm$^2$ (average of five samples), 0.17 µg cm$^{-2}$ for a piece of 14 cm$^2$ (based on one sample), and 0.11 ± 0.03 µg cm$^{-2}$ for a piece of 21 cm$^2$ (average of three samples). $^{14}$C was

analyzed on the pooled carbon collected from the five single filter pieces (with a total area of 35 cm$^2$) and the $F^{14}C$ was 0.54 ± 0.03. The concentration of EC on the sample filters varied from around 2 µg cm$^{-2}$ to 30 µg cm$^{-2}$. In most cases two filter pieces were combusted at the same time, but for the smallest samples three filter pieces were used, so that in all cases but one the blank is 5% or less of the sample amount. The concentration and $F^{14}C$ of TC on the blank filter were calculated by adding the

carbon concentrations of OC and EC and were 0.68 µg cm$^{-2}$ and 0.67, respectively.

The $^{14}$C values of all carbon fractions were blank corrected according to the mass balance equation:



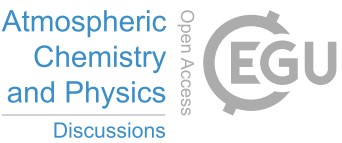

$$F^{14}C_S = \frac{F^{14}C_m \cdot M_m - F^{14}C_b \cdot M_b}{M_m - M_b},$$ (2)

where $F^{14}C_s$ is the fraction modern of the aerosol carbon collected on the filter, $F^{14}C_m$ and $M_m$ are the measured fraction modern and the measured concentration of the respective carbon fraction (TC, OC, WIOC, or EC), and $F^{14}C_b$ and $M_b$ the fraction modern and the concentration of the respective carbon fraction on the blank filter.

### 2.6 Measurements of sugars

The determination of carbohydrates is performed with an improved high performance anion-exchange chromatography with pulsed amperometric detection (HPAEC-PAD). This method is recommended for atmospherically relevant sugars (sugar alcohols, monosaccharides and monosaccharide anhydrides) and based on the method described by Iinuma et al., (2009). Before the analysis a filter aliquot (1x Ø10mm, 0.8 cm²) was eluted with 3 mL of ultra-pure water (Milli-QPlus, 185, Millipore), sonicated for 20 min and subsequenty centrifuged for 30 minutes. The extract was analyzed with an ion chromatography system (Dionex® ICS 3000), operated with a NaOH - gradient (initial concentration: 480 mM NaOH, final concentration: 650 mM NaOH) run on a Carbo Pac MA1 column. The quantification is carried out with external standards prepared from pure substances (Merck®, Fluka®).

### 2.7 Thermal-optical analysis

Filter pieces of 1.5 cm$^2$ were analyzed for OC and EC with a dual-optics Sunset Analyzer (Sunset Laboratory, Inc.) using the QUARTZ temperature protocol (Birch and Cary, 1996). In this instrument, filter samples are heated stepwise in a He-atmosphere to a maximum temperature of 840°C to volatilize organic material, then cooled down to 550°C and subsequently heated again stepwise to 870°C in a He-$O_2$ atmosphere to oxidize EC. All carbon gases are converted to $CH_4$ and detected with a flame ionization detector. As organic material partly pyrolyzes in the first part of the cycle, a laser beam is used to correct for charring by monitoring both transmission and reflection signals. The split between OC and EC is set when the transmission or reflection signals reach their initial values. EC values determined by both transmission and reflection method are compared later in the manuscript.

### 2.8 Estimation of OC, EC, WIOC concentrations

The OC combustion temperature of 360 °C in the ACS is likely not high enough to recover 100% of OC. Moreover, in the second heating step of the EC extraction procedure, a mixed fraction of OC and EC is removed from the filter to ensure the purity of the remaining EC. The concentrations of OC and EC extracted for $^{14}$C analysis ($M_{OC,e}$ and $M_{EC,e}$) are therefore lower than the actual concentrations of OC and EC in the aerosol sample ($M_{OC}$ and $M_{EC}$). If $F^{14}$C of TC is known, $M_{OC}$ and $M_{EC}$ can be estimated as follows:

$$M_{OC} = M_{TC} \frac{F^{14}C_{(TC)} - F^{14}C_{(EC)}}{F^{14}C_{(OC)} - F^{14}C_{(EC)}},$$ (3)





$$M_{EC} = M_{TC} - M_{OC,} \tag{4}$$

where $M_{OC}$, $M_{EC}$, and $M_{TC}$ are the mass concentrations of OC, EC, and TC, respectively; and $F^{14}C_{(OC)}$, $F^{14}C_{(EC)}$, and $F^{14}C_{(TC)}$ are the fraction modern of OC, EC and TC respectively. The recovery of OC and EC can consequently be estimated as $M_{OC,e}/M_{OC}$ and $M_{EC,e}/M_{EC}$.

In addition, we measured $^{14}C$ on WIOC. Estimating the concentration of WIOC is not straightforward, because the recovery of WIOC is unknown and cannot be derived in the same way as for OC. To estimate $M_{WIOC}$ we therefore assume two extreme cases: (1) WIOC is completely recovered, which most likely results in an underestimate of WIOC, and (2) WIOC shows the same recovery as OC, which is probably an overestimate, since WIOC is associated with more volatile primary organic

material and usually less WIOC is lost to charring. This results in a range of possible WIOC concentrations from a minimum of $M1_{WIOC}$ (complete recovery) to a maximum $M2_{WIOC}$ (recovery as OC).

### 2.9 Estimating source contributions to OC, EC, WIOC and WSOC

The $F^{14}C$ values of the different carbon fractions can be used to estimate source contributions to the

carbonaceous aerosol. For EC two main sources are considered: fossil fuel dominated sources ($EC_f$) and biomass burning ($EC_{bb}$), which in North-Western Europe is mainly from residential heating and fireplaces, since open wildfires are rare. $EC_{bb}$ can be estimated as:

$$EC_{bb} = M_{EC} \frac{F^{14}C_{(EC)} - F_f}{F_{bb} - F_f}, \tag{5}$$

where $F_{bb}$ stands for $F^{14}C$ of biomass burning emissions (see Table 1), and $F_f$ stands for the average

$F^{14}C$ of EC from other (fossil fuel dominated) combustion sources, such as natural gas and coal burning and emissions from mobile sources. In Europe, these sources emit predominantly fossil carbon and usually $F_f$ has been approximated as 0. However, in recent years an increasing fraction of biofuels has been added to gasoline and diesel used in road transport. This is expected to increase $F^{14}C$ in EC emitted from mobile emissions, although it is not clear how much biofuel addition affects $F^{14}C_{(EC)}$ (e.g.,

Bennett et al., 2008).

$EC_f$ can be calculated as:

$$EC_f = EC - EC_{bb} \tag{6}$$

Once $EC_{bb}$ is known, $OC_{bb}$ is calculated using an estimate of the primary OC/EC ratio of wood combustion emissions ($r_{bb}$; e.g., Bernardoni et al., 2013; Genberg et al., 2011; Szidat et al., 2006). This

is a very crude estimate, since $r_{bb}$ of domestic wood combustion in Europe is variable and not very well known. However, it is still useful to get an approximate idea of the relative contribution of primary biomass combustion vs. other modern carbon sources.



$$OC_{bb} = EC_{bb} \cdot r_{bb} \tag{7}$$

$OC_{bb}$ can be subtracted from the total OC and the remaining OC can be apportioned between OC from fossil dominated combustion and other modern OC. The latter consists of primary and secondary biogenic OC, but can also contain secondary OC from wood burning, especially in winter. This OC

fraction is therefore named $OC_{c,o}$ (OC contemporary, other) and can be calculated as:

$$OC_{c,o} = \frac{(F^{14}C_{(OC)}-F_f)\cdot M_{OC}+(F_f-F_{bb})\cdot OC_{bb}}{F_{bio}-F_f}, \tag{8}$$

where the fraction modern of $OC_{c,o}$ is approximated by $F_{bio}$, the fraction modern of biogenic OC. $OC_{c,o}$ can also contain small amounts of primary biogenic material, but in most cases the contributions to $PM_{2.5}$ are likely small. Other studies, have used $F_f \approx F_{fossil} = 0$ for the fossil fuel dominated combustion

sources, and OC emitted by biofuel combustion has been subsumed under $OC_{c,o}$. Currently in Europe, OC from biofuels is a very minor fraction of OC, but this might change in the future, if biofuels are used more extensively. In urban areas OC could also include a contribution of modern carbon from cooking sources (e.g., Allan et al., 2010; Schauer et al., 1996) however at a regional background site, this contribution is likely small.

Finally, OC from fossil dominated combustion sources ($OC_f$), can be estimated as:

$$OC_f = M_{OC} - OC_{c,o} - OC_{bb}. \tag{9}$$

To propagate experimental uncertainties and uncertainties in the parameters $F_f$, $F_{bb}$, and $r_{bb}$ to the final results, we conducted a Monte Carlo simulation with 10.000 individual calculations of $EC_{bb}$, $EC_{co}$, $OC_{bb}$, $OC_f$, and $OC_{c,o}$ using Eq. (3)-(7). For each calculation input values for OC and EC

concentrations, as well as $F^{14}C_{(OC)}$, and $F^{14}C_{(EC)}$ are chosen randomly from a normal distribution centered around the measured value with the experimental uncertainties as standard deviation. Similarly, random values for $F_f$, $F_{bb}$, and $r_{bb}$ are estimated as follows: For each of the parameters we estimated a central value and an upper and lower limit based on values reported in the literature. The random values are then chosen from a triangular frequency distribution, which has its maximum at the

central value and is 0 at the upper and lower limits. The central values, maxima and minima for $F_f$, $F_{bb}$, and $r_{bb}$ are reported in Table 1. Finally, the average and standard deviation of the 10.000 different estimates of $EC_{bb}$, $EC_{co}$, $OC_{bb}$, $OC_f$, and $OC_{c,o}$ are calculated. The average represents the best estimate of the concentration of each fraction and the standard deviation represents the uncertainty arising from uncertainties in the measurements and in $F_f$, $F_{bb}$, and $r_{bb}$.

We assume that $F_f$ lies between 0 (i.e. all combustion sources other than wood combustion are purely fossil) and 0.1 (modern carbon from biofuels contributes 10% to carbon emission from other combustion sources). The latter is a reasonable upper limit, since a contribution of biofuels to fuels in road transport of at least 5% was required by law in the Netherlands in 2011, but usually the contribution of modern carbon in the emitted particles is not equal to the contribution of modern carbon



in the fuels (e.g., Bennett et al., 2008, and references therein). The range for $F_{bb}$ is estimated as 1.1 to 1.2, with the most likely value of 1.15 based on (Genberg et al., 2011; Gilardoni et al., 2011; El Haddad et al., 2011; Minguillón et al., 2011; Szidat et al., 2006, 2007, 2009). Szidat et al. (2006) derive a best estimate of 6.25 for $r_{bb}$ from an extensive literature study of wood combustion emissions. A few

more recent studies give slightly lower values for $r_{bb}$, for example 5.3 for cooking fires (Christian et al., 2010), $3 \pm 2.4$ for typical wood stoves used in Austria (Schmidl et al., 2008) and 2.8 for log combustion in wood stoves (Zhang et al., 2013). Yttri et al., (2009) derive $r_{bb}$ of approximately 2.5 by ambient measurements in a location, where wood burning emissions dominate the concentrations of carbonaceous aerosol. Some larger values of $r_{bb}$ of more than 20 can also be found in the literature, but

these are often for special situations, such as strongly smoldering combustion (McMeeking et al., 2009), or burning of leaves and grasses, that are not representative for domestic wood combustion (Zhang et al., 2013). Moreover, there are strong indications that pellet and wood log burners have lower OC/BC ratios than traditional stoves (Heringa et al., 2012). Therefore we use a somewhat lower best estimate for $r_{bb}$ than Szidat et al., (2006) with a wider range from 3 to 7.

Since ratios of $WIOC_{bb}/EC_{bb}$ and $WSOC_{bb}/EC_{bb}$ are not often reported in the literature, it is not possible to separate biogenic and biomass burning carbon for these carbon fractions with any certainty. Therefore, the source apportionment for WIOC and WSOC considers only the fossil dominated combustion source discussed above, and a mixed biomass burning/biogenic "modern" source. First the concentration and $F^{14}C$ of WSOC are calculated as:

$$M_{WSOC} = M_{OC} - M_{WIOC}, \tag{10}$$

$$F^{14}C_{(WSOC)} = \frac{M_{OC} \cdot F^{14}C_{(OC)} - M_{WIOC} \cdot F^{14}C_{(WIOC)}}{M_{WSOC}}. \tag{11}$$

Since there is a range of possible values for $M_{WIOC}$, this uncertainty along with the experimental uncertainties in $F^{14}C$ and $M_{OC}$ are propagated to $M_{WSOC}$ and $F^{14}C_{(WSOC)}$ as a part of the Monte Carlo simulations described above. $M_{WIOC}$ is assumed to vary in the range of $M1_{WIOC}$ (complete recovery of

WIOC, see above) to $M2_{WIOC}$ (recovery as OC), with a most likely value at $M1_{WIOC} + 2/3 * (M2_{WIOC} - M1_{WIOC})$, since it is more likely that WIOC has a similar recovery as OC, than 100% recovery. However, since the recovery of OC is on average 75%, the results are not strongly sensitive to how the most likely value is chosen.

The concentration of WIOC from fossil dominated combustion sources is calculated as:

$$WIOC_f = WIOC \frac{F^{14}C_{(WIOC)} - F_c}{F_f - F_c}, \tag{12}$$

where $F_c$ is defined as $F^{14}C$ of the contemporary organic carbon (i.e. the sum of $OC_{bb}$ and $OC_{c,o}$) in the respective sample, estimated as:





$$F_c = \frac{OC_{bb} \cdot F_{bb} + OC_{bio} \cdot F_{bio}}{OC_{bb} + OC_{bio}} \tag{13}$$

Contemporary WIOC is calculated as $WIOC_c = WIOC - WIOC_f$. The concentrations of fossil fuel combustion WSOC, is calculated as $WSOC_f = OC_f - WIOC_f$ and contemporary WSOC as $WSOC_c = WSOC - WSOC_f$.

**3 Results**

**3.1 Method evaluation and quality control**

Several aspects of the combustion method, mainly focused on the separation of carbon fractions for [14]C
analysis, were thoroughly tested and evaluated by Dusek et al., (2014). For the current study, it was also necessary to evaluate the accuracy of the calculated TC, OC and EC concentrations based on Eq. (3) and (4), since these concentrations are the basis for further source apportionment of ambient aerosol. As basis for comparison we used OC and EC concentrations determined by thermal-optical analysis as described in section 2.8. TC concentrations were determined by thermal-optical analysis on
a subset of 18 filter samples. For eight of these filter samples the aerosol loadings were too high for reliable OC-EC analysis and therefore OC and EC concentrations are available only for 10 filter samples. For these 10 samples OC-EC concentrations estimated based on Eq. (3) and (4) could be compared with thermal optical analysis.

Figure 1 a shows that there is excellent agreement between TC determined by the ACS system and TC determined by the thermo-optical method (TC sunset). The slope is $1.07 \pm 0.03$, thus slightly higher than 1, which can be partly due to traces of water vapor or other impurities that are not removed entirely by the ACS method, but might also reflect a bias in the calibrated volume used for TC determination on the ACS.

Figure 1 b shows the ratio of calculated EC concentrations ($M_{EC}$) to EC determined by the thermo-optical method with transmission ($M_{EC,t}$) and with  reflection correction ($M_{EC,r}$) and using the average of $M_{EC,t}$ and $M_{EC,r}$ The data points correspond to individual filter samples and the red triangles are the average ratios for all filter samples. $M_{EC}$ is on average roughly 10% higher than $M_{EC,t}$ and 10% lower
than $M_{EC,r}$. The standard deviation is approximately 25% of the average ratio, which is comparable to the precision of different thermo-optical methods for EC (e.g., Schmid et al., 2001). We can therefore conclude that our method of estimating EC and OC concentrations gives comparable results to well-established methods for measuring EC and OC concentrations.

The recovery of OC and EC, estimated according to section 2.8, is on average 75% for OC and 82% for EC. For one sample the recovery of OC was calculated as 178%, likely due to an unrealistically low $^{14}C_{(TC)}$ value. This sample was excluded from further analysis and was also not considered in Figure 1b.


For the subset of samples displayed in Figure 1b, the average recovery of EC was 83%, compared to the recovery with respect to thermo-optical analysis $M_{EC,e}/M_{EC,av}$ of 79 %.

In $^{14}$C source apportionment, biomass burning is usually considered the main source of contemporary
EC. However, there is evidence that primary biogenic particles, such as pollen fragments, can be thermally very refractory and be combusted together with EC (Wittmaack, 2005). Whereas pollen themselves are usually not found in PM2.5, the fragments of ruptured pollen are in the sub-micrometer size range (Taylor et al., 2004) and can potentially contribute modern carbon to the EC fraction. Therefore, $F^{14}C_{(EC)}$ was evaluated against levoglucosan, which is another tracer of biomass burning.
Figure 2 shows that for most samples there is a clear correlation between the fraction of levoglucosan in TC and $F^{14}C_{(EC)}$. However, there are three samples with strongly elevated $F^{14}C_{(EC)}$ but low levels of levoglucosan (shown in red). These samples were taken in spring 2014 during a time period with reported high pollen concentrations. In this period, many surfaces were covered with a fine yellow dust. This dust was strongly visible on the PM2.5 impactor plate of the High-Volume sampler and also
present on the filter itself.  For these three samples glucose and sucrose concentrations, which are tracers for primary biological material, were strongly elevated. Glucose/TC levels were 18 ng µg$^{-1}$ compared to an average of 1.7 ng µg$^{-1}$ for the rest of the year and sucrose/TC levels were 81 ng µg$^{-1}$ compared to an average of 11 ng µg$^{-1}$ for the rest of the year. These observations provide the first evidence that primary biological material can be responsible for elevated $F^{14}C_{(EC)}$ during spring.
Weather this contribution comes from strong charring or the fact that a part of the primary biological material survives the thermal treatment designed to purify EC is not entirely clear. For further calculations in the manuscript, the measured $F^{14}C_{(EC)}$ of the three samples marked in red was replaced by $F^{14}C_{(EC)}$ calculated based on measured levoglucosan concentration and the regression line in figure 2. The highly refractory part of OC that was apparently incorrectly classified as EC, was therefore
disregarded.

The offset of the regression line in Figure 2 is not 0, which indicates that there is contemporary carbon in EC that is not related to biomass burning. One main reason is the addition of biofuel to diesel and gasoline, which in the Netherlands must currently contain at least 5% biofuel. In addition, cooking
could contribute to contemporary EC, though this contribution is relatively uncertain. An increase in $F^{14}C_{(EC)}$ of roughly 0.03 to 0.05 can be expected from charring of organic compounds (Dusek et al., 2014; Zhang et al., 2013). A bias of 0.04 was therefore subtracted from $F^{14}C_{(EC)}$. In addition there is evidence that levoglucosan is not a stable tracer, but can be degraded by photochemical reactions during long-range transport (Hoffmann et al., 2010). This means that the offset in $F^{14}C_{(EC)}$ could also
partially be caused by underestimated levoglucosan concentrations.

### 3.2 Source apportionment in different seasons and air mass conditions

We analyzed $F^{14}C$ in TC, OC, WIOC and EC fractions for 26 PM2.5 samples, taken in 2011 and 2012.
Four-day air mass back trajectories were calculated with the HYSPLIT model every 24 hours for each





sample from the sampling start time to the sample end time. Most filter samples can be associated with one of two main source regimes: Continental or modified marine and are shown in Figure 3. The air mass back trajectories for the filters classified as "continental" are shown as green lines. The trajectories originate mostly to the East or South of the Netherlands and spend considerable time over

land. A few trajectories are also from the West, because occasionally a filter sample could not be stopped in time before a change of wind direction. However, for each continental filter sample such trajectories accounted only for a small fraction of the total sampling time. The air mass back trajectories for filter samples classified as "modified marine" originate over the Atlantic and spend only a brief time over land before reaching the Netherlands, mostly over Great Britain, Belgium. These

trajectories are shown in red. The trajectories colored in blue correspond to the cleanest air mass conditions, when the air arrived from the North and did not cross any land masses before reaching the Netherlands. However, there were only two filter samples with these air mass conditions, and they were therefore included into the modified marine cases.

Since the concentration of carbonaceous aerosol over the ocean is low, the aerosol samples in the modified marine case are likely dominated by regional emissions from in and around the Netherlands. The majority of the particles in these samples had been emitted within the last 24 hours and the aerosol carbon is therefore characterized by a short aging time. These samples will be referred to as regional pollution (reg) samples hereafter. In the continental case long-range transport contributes to the

carbonaceous aerosol in addition to the regional emissions. The continental samples also contain particles that were emitted several days earlier, which results in a longer average atmospheric residence time of the collected particles. Due to the longer residence times the aerosol carbon in the continental samples is on average also more aged.

Table 2 shows $F^{14}C$ for the different seasons and air mass types. The number of samples averaged for each case is given in column 2. Each of the samples was collected over several days and the averages given in Table 2 cover 23 days for the spring samples, 39 days for the summer samples, 31 days for the fall samples, and 25 days for the winter samples. The average $F^{14}C_{(EC)}$ in spring excludes the three EC samples that were influenced by primary biogenic material and therefore only three samples are

considered. For all seasons EC is dominated by fossil sources and OC by modern sources. Water insoluble OC has a considerably lower $F^{14}C$ than OC, which clearly indicates a higher contribution of fossil sources to WIOC.
$F^{14}C$ of all carbon fractions is lowest in summer, which is somewhat surprising since the production of biogenic SOA from contemporary precursors should be high in spring and summer and should increase

$F^{14}C$ of OC and TC compared to fall and winter. The summer of 2011 was relatively cold and had the second highest amount of precipitation of recorded history with 389 mm averaged over the Netherlands compared to a climatological mean of 219 mm. These weather conditions could have suppressed SOA formation and therefore $F^{14}C$ for this summer might be biased low. However, two samples taken in the summer of 2012 in the course of a field experiment, showed similar $F^{14}C$ values as the samples in this

study, with $F^{14}C_{(TC)}$ around 0.55.



$F^{14}C_{(OC)}$ remained relatively constant for all seasons and air mass conditions, but was highest in spring. $F^{14}C_{(EC)}$ varied more strongly and was low in summer and high in winter, likely caused by an increase in biomass combustion activities in winter. $F^{14}C_{(EC)}$ was low in air masses with regional pollution, even

in winter, and much higher in air masses with continental air mass origin.

The $F^{14}C$ values were used as input for source apportionment calculations to estimate OC and EC from biomass burning ($OC_{bb}$ and $EC_{bb}$), OC and EC from fossil dominated sources ($OC_f$ and $EC_f$) and other contemporary ($OC_{c,o}$) with the Monte Carlo approach described in section 2.9. Figure 4 shows a typical

frequency distribution of $OC_f$, $OC_{bb}$, and $OC_{c,o}$ values, resulting from 10 000 individual calculations, each with randomly chosen input parameters from the ranges given in section 2.9. These frequency distributions indicate the range and probability of possible values for each carbon fraction, taking into account all uncertainties in measurements and assumptions.

$OC_f$ was constrained in a relatively narrow range, whereas estimates for $OC_{bb}$ and $OC_{c,o}$ varied over a much wider range reflecting the large uncertainty in $r_{bb}$. The individual estimates of $OC_{bb}$ and $OC_{c,o}$ are not independent. Since the sum of $OC_{bb}$ and $OC_{c,o}$ is narrowly constrained as OC - $OC_f$, large values of $OC_{bb}$ correspond to small values of $OC_{c,o}$ and vice versa. Despite the considerable uncertainty in $OC_{bb}$ and $OC_{c,o}$ some conclusions can be drawn from the distributions in Figure 4, for example that $OC_{c,o}$

was the most abundant of the three components and that both $OC_{bb}$ and $OC_{c,o}$ were clearly present in the sample.

The results of the source apportionment calculations are summarized in Table 3. The values represent averages for different seasons and air mass conditions and the corresponding standard deviations. The

sum of $OC_{bb}$ and $OC_{c,o}$ is reported as contemporary OC ($OC_c$). The seasonal variation is characterized by low concentrations in summer, which can be attributed partially to rainy conditions and probably also to higher planetary boundary layers in summer. The main sources of fossil elemental carbon ($EC_f$) in the Netherlands do not show a strong seasonal variation and its concentrations should therefore be relatively constant throughout the year. However, there are relatively high $EC_f$ concentrations in fall

and the concentrations of all other carbon fractions are elevated as well. This might be caused by easterly air masses that often contain higher levels of pollution than westerly ones from the sea and occur more frequently during fall. $OC_c$ and $WSOC_c$ are elevated in spring 2011 and $OC_{bb}$ and $EC_{bb}$ are high in winter and low in spring and summer 2011. Least variable is $WSOC_f$, which has relatively higher concentrations in summer than the other carbon fractions. The standard deviations reflect the

variability of pollution levels, which are generally higher in fall and winter. In summary, there are some indications of a seasonal variation in carbon concentrations, but the variability within each season is high and strongly influenced by weather and air mass conditions.

Comparing the concentrations for different air mass conditions shows clearer differences. For regional

pollution carbon concentrations are rather low and less variable than seasonal averages. If we assume





that aerosol concentrations from long-range transport are superimposed on the regional background, then the increase in aerosol concentrations during continental air mass conditions gives an indication of the importance of long-range transport for the carbonaceous aerosol concentrations in the Netherlands. Carbon fractions that only show a small increase under continental air mass conditions are mostly of

regional origin, whereas for carbon fractions that increase more strongly, regional sources are less prominent.

The concentrations of most carbon fractions increase by a factor of 3.4 – 3.6 in continental compared to regional air masses. Notable exceptions are fossil EC and fossil WIOC, which only increase by a factor

of approximately 2.5. The biggest source of fossil EC and fossil WIOC are traffic emissions. Our data therefore indicate that the regional contribution in the Netherlands is relatively strong for OC and EC from traffic sources and the influence of long-range transport less important. On the other hand, $EC_{bb}$, $WIOC_c$, and $WSOC_f$ increase by more than a factor of 4 under continental air mass conditions. Especially concentrations of $EC_{bb}$ are very low in regional pollution, suggesting that most of $EC_{bb}$

originates from outside the Netherlands. Since primary OC from fossil sources is usually water insoluble, $WSOC_f$ is most likely of secondary origin. Its concentration increases considerably in continental air masses that contain emissions from more distant sources after a longer aging time in the atmosphere. $WIOC_c$ and its potential origin will be discussed in more detail below.

Differences in precipitation are a first order effect that could explain part of the difference between carbon concentrations during regional and continental air masses conditions. Usually precipitation is higher under westerly wind directions (i.e. marine modified) than under the easterly wind directions that are associated with continental air mass conditions. Wet removal of aerosol strongly lowers aerosol concentrations during precipitation events. Therefore we compared the average duration and amount of

precipitation measured at the Cesar Observatory during the sampling times of filters representative for regional and continental air mass conditions. The highest TC concentrations above 5 µg m$^{-3}$ occur only on dry days, however on dry days concentrations below 2 µg m$^{-3}$ are also measured frequently. The rainfall duration was on average 1 hr/day for continental conditions and 2 hrs/day for regional conditions and the amount was 1.2 vs 3.4 mm/day. Even though there is on average more precipitation

under regional air mass conditions, one additional hour of rain per day is unlikely to reduce aerosol concentrations by a factor of 3. Furthermore the ratios continental/regional are different for $EC_{bb}$ and $EC_f$ and it is not likely that these compounds would be scavenged differently. Therefore, the influence of precipitation is unlikely to be the main reason for the difference in carbonaceous aerosol concentrations between air mass conditions. Also the wind speed is not much lower during continental

air mass conditions. The wind speeds averaged around 3.5 m/s under continental air mass conditions and 4.5 m/s under regional air mass conditions. Despite the fact that higher wind speeds are usually associated with a higher boundary layer and more diluted pollutants, there was not a very clear relationship between TC concentrations and wind speed. For example the highest TC concentrations of 9.5 µg m$^{-3}$ were measured in a period with relatively low wind average speed (2.9 m/s), but almost

equally high concentrations of 8.2 µg m$^{-3}$ were measured in a period with above average wind speeds of



4.4 m/s. The three lowest TC concentrations (below 1 µg m$^{-3}$) were measured during time periods with average wind speeds (3.7 m/s), high wind speeds (8.3) m/s, and slightly elevated wind speeds (4.8 m/s). Therefore we think it unlikely that meteorological conditions are mainly responsible for the differences in carbon concentrations under regional and continental air mass conditions.

Figure 5 shows the concentrations of OC and EC apportioned to the various sources averaged over different seasons as colored bars and the concentrations corresponding to the individual filter samples as black dots. The red lines show the medians for each season. The error bars show uncertainties of the average calculated by propagating the uncertainties of the individual data points, obtained by the Monte

Carlo simulation. These error bars therefore reflect the methodological uncertainties rather than variability of the data. These methodological uncertainties are smaller than the standard deviation of the individual measurement points that were reported in Table 3. The strong variability in the concentrations of EC and OC fractions therefore primarily reflect the actual differences in atmospheric concentrations and not the uncertainties of our measurements and assumptions.

Figure 5 a shows that the high average concentration of EC$_f$ in fall is mainly due to three pollution events. A similar pollution event also occurred in winter. This example shows the importance of occasional pollution events for aerosol concentrations in the Netherlands. Constructing seasonal averages therefore requires a large number of samples that also capture typical frequencies of pollution

events. This would require sampling over several years and careful selection of samples to represent average meteorological conditions and air mass origin for the respective season. This was not possible for this study and therefore the seasonal variation should not be over-interpreted.
The medians are less influenced by outliers and most carbon fractions in Figure 5 (a) and (b) show a much weaker seasonal variation in the median than in the mean. However, the median OC$_{c,o}$ is still

clearly elevated in spring. The two samples with very high OC$_{c,o}$ concentrations correspond to two of the three pollen events. EC$_{bb}$ and OC$_{bb}$ show also clear seasonal variation with low median concentrations in spring and summer, when domestic biomass combustion is not a major source of aerosol carbon in Western Europe.

Figure 6 shows the concentrations of OC and EC during regional and continental air mass conditions.

Pollution episodes characterized by high carbon concentrations occur only during continental air mass conditions. When pollutants originate in and around the Netherlands, the concentrations of EC and OC are less variable and rather low. However, carbon concentrations within this low range also occur regularly under continental air mass conditions, suggesting that continental air masses do not always carry high aerosol loadings to the Netherlands. The concentrations under continental air mass

conditions are not correlated with precipitation or wind speed, and therefore likely depend on the air mass origin.

EC$_{bb}$ and OC$_{bb}$ are low in regional pollution, even in winter, suggesting low biomass combustion emission from within the Netherlands. Under continental air mass conditions carbon concentrations



from biomass combustion are elevated during fall and winter. Very low concentrations of $EC_{bb}$ and $OC_{bb}$ occur in summer and spring, when residential biomass combustion is low throughout Europe.

The higher average concentrations of fossil carbon in continental air masses are mainly caused by pollution events and the median concentrations are comparable. On the other hand the median

concentrations of $OC_{c,o}$ are clearly higher under continental than under regional air mass conditions.

**3.3 Source apportionment of water-soluble and water insoluble OC**

Figure 7 (a) shows the contribution of $WSOC_f$, $WIOC_f$, as well as $WSOC_c$, and $WIOC_c$ to the total organic carbon in different seasons during 2011/12. The fraction of WSOC is the sum of the blue areas, which account for 2/3 ~ 3/4 of the total OC. The contribution of WSOC to OC is somewhat higher in

spring and summer than in fall and winter. Throughout the year, WSOC is dominated by contemporary carbon, which reflects that the two main sources of contemporary OC, namely biomass combustion and biogenic SOA, are largely water-soluble. WIOC consists to roughly equal parts of fossil and contemporary carbon with slightly higher fossil contributions in summer and slightly higher contemporary contributions in fall and winter.

The contributions of fossil and contemporary carbon fractions to OC (Figure 7b) do not change strongly for different air mass origins, even though the absolute concentrations of OC increased strongly in continental air masses.

The sources of $WIOC_c$, which contributes up to a quarter of the total OC in this study are currently not very well known. Figure 8 shows a positive correlation between the concentrations of $WIOC_c$ and

$EC_{bb}$. The three points marked in red correspond to the pollen events, where high $F^{14}C_{(EC)}$ values were caused by refractory primary biogenic aerosol. $EC_{bb}$ concentrations were corrected for this effect as described in section 3.1. $WIOC_c$ is high for the three samples collected during pollen events, which shows that primary biological material can contribute significantly to the water insoluble carbon.

A linear least squares fit to the data, excluding the three highest data points and the pollen events, an

intercept of $0.10 \pm 0.02$ and an $R^2$ of 0.74. The offset is small, corresponding to approximately 0.1 mg/m$^3$ of $WIOC_c$ that is independent of $EC_{bb}$. In other words, most of the contemporary WIOC in the Netherlands seems to be associated with biomass combustion. There is no strong evidence of a major contribution of biogenic SOA to WIOC, which should be highest in spring and summer, when $EC_{bb}$ and $WIOC_c$ concentrations are both small.

Figure 9 shows a scatter plot of $WIOC_f$ and $EC_f$ concentrations. $EC_f$ is emitted by the combustion of fossil fuels and a positive correlation of $WIOC_f$ with $EC_f$ implies that primary emissions by fossil fuel combustion is an important source for $WIOC_f$ as well. There is one outlier data point, for which it is possible that the EC concentration calculated by Equations (2) and (3) is overestimated. A linear regression excluding the four data points with highest $EC_f$ concentrations has a slope of 0.32, an

intercept of 0.1, and an $R^2 = 0.68$. If the three highest data points are included the slope is: 0.79, the





intercept -0.045, and the $R^2 = 0.92$, but it is generally not good practice to fit such bimodal data with a linear regression. The low intercept indicates that primary emissions from fossil fuel combustion is the predominant source of fossil WIOC. The slope of such a regression line can give real world constraints on OC/EC emissions ratios of fossil sources, in this case we can suggest that OC/EC ratios of an

integrated fossil source should be below 0.8.

In the lower concentration range there is considerable scatter in the data points and individual $WIOC_f/EC_f$ are in the range of 0.4 to 1. However, compared to the typical variability of ambient OC/EC ratios, which can range from $1 - 10$, or even higher in remote locations ( e.g., Khan et al., 2016; Sandrini et al., 2014 and references therein) or even of $OC_f/EC_f$ ratios, this range of variability is

relatively low. The values of $WIOC_f/EC_f$ measured in this study are comparable to OC/EC ratios of primary fossil emissions, such as vehicular traffic in tunnel studies (e.g., Chirico et al., 2011). OC/EC emission ratios of coal combustion can be in the same range or higher (e.g., Chen et al., 2015). Coal combustion is a negligible source in and around the Netherlands, however it might be important during long-range transport events of pollution from Eastern Europe.  $WIOC_f/EC_f$ values measured during this

study were overall higher in winter than in the other seasons.

Average $WIOC_f/EC_f$ ratios are similar under regional pollution ($0.6 \pm 0.2$) and under continental air mass conditions ($0.6 \pm 0.3$). This indicates that $WIOC_f/EC_f$ ratios might stem from similar sources in both cases. The $WIOC_f/EC_f$ ratio is $0.6 \pm 0.3$ averaged over all samples, which might be indicative for an integrated primary OC/EC emission ratio for fossil sources in western Europe.

$WSOC_f$ is usually thought to result mainly from secondary formation. The ratio of $WSOC_f/EC_f$ is relatively low under regional air mass conditions, which sample relatively fresh emissions. It increases under continental air mass conditions, where older and more processed aerosol is sampled. However, the variability of the $WSOC_f/EC_f$ ratios is large, indicating that $WSOC_f$ and $EC_f$ do not originate from a common source. The average ratios in continental and regional air mass conditions are not significantly

different.

### 4. Discussion: Aerosol in the Netherlands compared to other regions

The average $F^{14}C_{(TC)}$ measured in the Netherlands over the course of a year is approximately 0.65. This is in the range of $F^{14}C_{(TC)}$ measured at different locations throughout Europe, roughly 0.5 to 0.85, depending on location and season (Heal, 2014, and references therein). $F^{14}C_{(TC)}$ values are usually at

the higher end of the range in rural locations and at the lower end in urban areas. $F^{14}C_{(TC)}$ at our polluted regional background site lies roughly in the middle of this range, corresponding to expectations. $F^{14}C_{(TC)}$ values in this study were slightly lower in conditions dominated by regional pollution (0.6) than in conditions influenced by long-range transport (0.7). This indicates that aerosol carbon in the Netherlands is more strongly influenced by fossil sources than aerosol carbon transported

from southern and central Europe. In the Unites States $F^{14}C_{(TC)}$ values for sub-urban and rural sites are usually higher ($0.7 - 1$; Lewis and Stiles, 2006; Lewis et al., 2004; Schichtel et al., 2008). In Japan



$F^{14}C_{(TC)}$ is usually lower and ranged from 0.3 – 0.5 in studies conducted mostly in urban or suburban areas (Fushimi et al., 2011; Heal, 2014, and references therein; Morino et al., 2010). In China $F^{14}C_{(TC)}$ can vary strongly by location from predominantly fossil in large urban centers (Zhang et al., 2015) to mainly biogenic in some rural areas (Zhang et al., 2014).

In this study $F^{14}C_{(TC)}$ was highest in spring and lowest in summer. There are not many studies, where data were compared for the same site in different seasons, and most of them analyzed only a few samples per season, making them vulnerable for sampling bias, as discussed above for this study. However, a general survey seems to indicate that for sites situated in cleaner regions, such as Scandinavia (Genberg et al., 2011; Szidat et al., 2009; Yttri et al., 2011), rural Spain (Minguillón et al.,

2011), the rural US (Bench, 2004; Schichtel et al., 2008) and several other rural background sites throughout Europe  (Gelencsér et al., 2007), $F^{14}C_{(TC)}$ was lower in winter than in summer. In these locations, biogenic SOA presumably contributed significantly to the total aerosol concentrations in summer. In more polluted locations, such as the Po Valley (Gilardoni et al., 2011), Barcelona (Minguillón et al., 2011) and Tokyo (Minoura et al., 2012), $F^{14}C_{(TC)}$ tended to be higher in winter than

in summer.  In Birmingham (Heal et al., 2011), there was no significant seasonal variation in $F^{14}C_{(TC)}$. The results of our study are more in line with the observations at polluted sites, which seems reasonable given the high population and traffic density of the surrounding urban areas. However, the results of our study might also be influenced by the unusually high amount of precipitation and low solar radiation during the summer of 2011.

$F^{14}C_{(OC)}$ was around 0.8, which is on the higher end of several studies conducted in Europe, where $F^{14}C_{(OC)}$ varied between 0.55 and 0.8 (Ceburnis et al., 2011; Dusek et al., 2013; Heal et al., 2011; Minguillón et al., 2011; Szidat et al., 2006). However, most of these values were measured in urban areas and it is likely that $F^{14}C_{(OC)}$ is higher than that range at rural or regional sites. $F^{14}C_{(OC)}$ did not vary strongly between air mass conditions and seasons in our study, and also other measurements in

Europe have shown no or weak variation between the seasons (Heal, 2014, and references therein).

If the samples influenced by primary biogenic material are disregarded, the average $F^{14}C_{(EC)}$ measured at the Cesar site is 0.2, with on average 0.15 in summer and 0.25 in winter. This is in the range of or slightly higher than $F^{14}C_{(EC)}$ values reported for several European cities, such as Zürich (Szidat et al., 2006), Birmingham (Heal et al., 2011), Barcelona (Minguillón et al., 2011), or Göteborg (Szidat et al.,

2009), but lower than $F^{14}C_{(EC)}$ measured in several background regions, especially in winter, such as ~ 0.4 in rural Sweden (Szidat et al., 2009), ~ 0.3 in rural Spain (Minguillón et al., 2011), and 0.3 to 0.8 during winter-time smog episodes in Switzerland (Zotter et al., 2014b). In addition to the limited number of studies that directly measured $F^{14}C_{(EC)}$ for PM2.5 in Europe, several studies estimated the contribution of biomass burning to EC based on $F^{14}C_{(TC)}$ and other tracers, such as levoglucosan.  In

comparison with these studies we can conclude that the winter-time contribution of biomass burning to EC in the Netherlands is lower than in many other regions in Europe (e.g., Genberg et al., 2011; Gilardoni et al., 2011; Yttri et al., 2009). Additionally, our study shows that $F^{14}C_{(EC)}$ increases from 0.2



under regional pollution to 0.3 under continental air mass conditions. This is consistent with long-range transport of carbonaceous aerosol from other regions in Europe with higher $F^{14}C_{(EC)}$.

Relatively few studies determined $F^{14}C_{(WIOC)}$. In several locations in Europe WIOC contained roughly equal fractions of contemporary and fossil carbon (Szidat et al., 2007, 2009), comparable to our study.

This is however not necessarily the case in other regions of the world. In Mexico city, where OC is dominated by fossil emissions (Aiken et al., 2010), WIOC is also mainly fossil. At a regional background site in South-East China, where biomass burning is a strong source, WIOC is mainly modern (Zhang et al., 2014).

$WIOC_f/EC_f$ ratios determined in Switzerland or Sweden were comparable to our values ranging from

roughly 0.5 - 1 (Szidat et al., 2004, 2009); the ratios were not directly reported but estimated from graphs in these publications. Two studies done in China report higher $WIOC_f/EC_f$ ratios of around 1 (Zhang et al., 2014) and in the range from 1 – 2.5 in southeast China (Liu et al., 2013). These high values could be explained by less efficient combustion in an older vehicle fleet and also higher OC/EC emission ratios from coal burning that is much more common in China than in western Europe.

In contrast, WSOC is dominated by modern sources in all regions of the globe with usually only 0 – 20% contributions from fossil sources (e.g., Kirillova et al., 2010, 2013, 2014; Szidat et al., 2006, 2009; Wozniak et al., 2012) This reflects that the main sources of modern OC, biomass burning and SOA formation, produce largely water soluble carbon. The data from the Cesar site fall in this range with a fossil fraction of WSOC below 0.2 (Fig. 7).

We also determined the ratio of water soluble to water insoluble fossil carbon $(WSOC/WIOC)_f$. This ratio is higher in spring and summer than in winter (1.2 vs. 0.6). It is also higher in aged continental air mass conditions than in regional pollution (1.2 vs. 0.7). Szidat et al., (2009) also noticed that the fractional contribution of WSOC to fossil OC increases in summer in Scandinavia. This is consistent with a secondary origin of $WSOC_f$. SOA formation is usually enhanced in spring and summer due to

higher solar radiation. Longer atmospheric aging times of primary emissions usually lead to higher SOA concentrations. If $WSOC_f$ is regarded as a proxy for $SOA_f$, then the above $(WSOC/WIOC)_f$ ratios indicate that secondary formation contributes rough 50% to fossil OC in spring and summer versus 40% in winter; and 50% in continental air masses versus approximately 30% in regional pollution. Much higher contributions of $SOA_f$ (around 70%) to fossil OC have been found in south China (Zhang

et al., 2014). Zotter et al. (2014b) observed that $SOA_f$ concentrations are larger in the Los Angeles Area than in European cities, in contrary to $EC_f$ concentrations, which are higher in Europe. They attributed this to the higher fraction of diesel cars in Europe, which emit a larger fraction of EC and less SOA precursors than gasoline cars. Fushimi et al. (2010) concluded that in suburban Tokyo, fossil sources make a dominant contribution to the very high SOA concentrations observed in daytime. SOA

concentrations in that study were up to 10 times higher than primary OC concentrations, so that fossil OA was predominantly of secondary origin. Even though secondary formation contributes significantly





to fossil OA in the Netherlands, compared to other, more photochemically active regions the contribution is relatively moderate.

**Conclusions**

This study presents source apportionment of carbonaceous aerosol at a regional site in the Netherlands
over the course of a whole year. Radiocarbon was measured in several carbon fractions of PM2.5, namely total carbon (TC), organic carbon (OC), water insoluble carbon (WIOC) and elemental carbon (EC). $F^{14}C_{(EC)}$ is strongly correlated with the fraction of levoglucosan in TC, except for three samples in spring time, which were collected under very high pollen concentrations. Our results suggest that refractory primary biogenic material can survive the OC removal steps and bias $F^{14}C_{(EC)}$ to higher
values.

$F^{14}C$ values of all carbon fractions are on average lower for regional pollution, which indicates that aerosol carbon in the Netherlands contains a higher contribution from fossil sources than the aerosol transported to the Netherlands. Overall, the highest $F^{14}C$ values were measured in spring and the lowest $F^{14}C$ values in summer. However, there are only a few data points per season, so specific
meteorological conditions, such as the very wet summer of 2011 may have strongly affected the differences between seasons, which may not be representative for a multi-year mean.

The concentrations of all carbonaceous fractions were on average higher in continental air masses influenced by long-range transport than for regional pollution. The difference in TC concentrations was on average a factor of 3. However, the average concentrations in continental air masses were strongly
influenced by a few pollution episodes with concentrations much higher than for regional pollution. This highlights the importance of long-range transport that episodically caused high TC concentrations in the Netherlands. Fossil EC and fossil WIOC concentrations increased less than TC in continental air masses. We conclude that these carbon fractions have a larger relative contribution from regional sources than TC.  Carbonaceous aerosol from biomass burning and fossil WSOC increased strongly in
continental air masses, suggesting major sources outside the Netherlands. Fossil WSOC can be attributed mainly to secondary formation and the longer aging times during long-range transport lead to higher concentrations of secondary carbon.

Fossil WIOC/EC was on average 0.6, well within the range of primary OC/EC ratios from vehicular emissions. This ratio did not change between regional and continental air masses. One of the most
interesting results of our study is that, even though a large fraction of carbon emitted by biomass burning is water soluble, long-range transport of biomass smoke is the most important source of $WIOC_c$ in the Netherlands. However, it is not clear if this holds in other regions of the world and especially for other particle size ranges, such PM10, or TSP. In these size ranges primary biogenic material can also contribute to $WIOC_c$.

**Acknowledgments:**





This work was funded by the Dutch science foundation (NWO, grant Nr. 820.01.001). We would like to thank Dicky van Zonneveld, Henk Been, and Anita Aerts-Bijma for their [14]C analysis work. We gratefully acknowledge the work of Mattia Monaco and Arthur Kappetin, who contributed to sampling and analysis as part of their Master projects.

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



Table 1. The range of input values for crucial parameters used in the Monte Carlo simulations: The fraction modern of biomass burning aerosol ($F_{bb}$), the fraction modern of fossil dominated sources ($F_f$), the fraction modern of biogenic aerosol ($F_{bio}$) and the OC/EC ratio of domestic biomass combustion ($r_{bb}$).

|          | min. | central value | max. |
|----------|------|---------------|------|
| $F_{bb}$  | 1.1  | 1.15          | 1.2  |
| $F_f$     | 0    | 0.07          | 0.1  |
| $F_{bio}$ | -    | 1.04          | -    |
| $r_{bb}$  | 3    | 5             | 7    |





Table 2: Average fraction modern ($F^{14}C$) after blank correction ± standard deviations of $F^{14}C$ determination for total carbon (TC), organic carbon (OC), elemental carbon (EC) and water insoluble organic carbon (WIOC) in different seasons and for regional (reg) and continental (co) air mass origin. The second column gives the number of samples averaged for each case.

|  | Samples | $F^{14}C_{(TC)}$ | $F^{14}C_{(OC)}$ | $F^{14}C_{(WIOC)}$ | $F^{14}C_{(EC)}$ |
|---|---|---|---|---|---|
| winter | n = 5 | 0.664 ± 0.068 | 0.742 ± 0.071 | 0.529 ± 0.093 | 0.262 ± 0.092 |
| spring | n = 6 | 0.760 ± 0.093 | 0.886 ± 0.052 | 0.660 ± 0.173 | 0.173 ± 0.095 |
| summer | n = 6 | 0.574 ± 0.073 | 0.713 ± 0.052 | 0.480 ± 0.127 | 0.157 ± 0.042 |
| fall | n = 7 | 0.627 ± 0.100 | 0.782 ± 0.058 | 0.595 ± 0.097 | 0.227 ± 0.079 |
| reg | n = 13 | 0.618 ± 0.065 | 0.766 ± 0.091 | 0.509 ± 0.088 | 0.170 ± 0.050 |
| co | n =11 | 0.698 ± 0.135 | 0.810 ± 0.082 | 0.637 ± 0.152 | 0.307 ± 0.107 |





Table 3: Average concentrations of OC, EC, WIOC an WSOC and the respective contribution of contemporary and of fossil-dominated sources for different seasons and for regional (reg) and continental (co) air mass conditions.

| | Winter $C_{av}$ [µg m$^{-3}$] | spring $C_{av}$ [µg m$^{-3}$] | summer $C_{av}$ [µg m$^{-3}$] | fall $C_{av}$ [µg m$^{-3}$] | reg $C_{av}$ [µg m$^{-3}$] | co $C_{av}$ [µg m$^{-3}$] |
|---|---|---|---|---|---|---|
| TC | 2.7 ± 3.0 | 3.6 ± 2.2 | 1.4 ± 0.5 | 4.1 ± 3.2 | 1.4 ± 0.7 | 4.9 ± 3.7 |
| OC | 2.1 ± 2.3 | 3.0 ± 2.0 | 1.0 ± 0.3 | 3.0 ± 2.3 | 1.0 ± 0.5 | 3.8 ± 2.9 |
| OC$_f$ | 0.69 ± 0.78 | 0.46 ± 0.31 | 0.35 ± 0.11 | 0.88 ± 0.71 | 0.28 ± 0.10 | 1.0 ± 0.7 |
| OC$_c$ | 1.4 ± 1.5 | 2.5 ± 1.6 | 0.68 ± 0.24 | 2.1 ± 1.6 | 0.76 ± 0.43 | 2.8 ± 2.1 |
| EC | 0.66 ± 0.72 | 0.56 ± 0.22 | 0.34 ± 0.16 | 1.2 ± 0.92 | 0.38 ± 0.18 | 1.1 ± 0.9 |
| EC$_f$ | 0.51 ± 0.51 | 0.52 ± 0.17 | 0.33 ± 0.15 | 1.0 ± 0.78 | 0.35 ± 0.16 | 0.93 ± 0.69 |
| EC$_{bb}$ | 0.15 ± 0.22 | 0.04 ± 0.05 | 0.02 ± 0.01 | 0.14 ± 0.15 | 0.02 ± 0.02 | 0.16 ± 0.18 |
| WIOC | 0.80 ± 0.88 | 0.63 ± 0.29 | 0.29 ± 0.12 | 1.2 ± 1.0 | 0.35 ± 0.15 | 1.2 ± 1.0 |
| WIOC$_f$ | 0.44 ± 0.50 | 0.23 ± 0.07 | 0.17 ± 0.07 | 0.52 ± 0.36 | 0.19 ± 0.07 | 0.52 ± 0.43 |
| WIOC$_c$ | 0.37 ± 0.38 | 0.41 ± 0.22 | 0.12 ± 0.05 | 0.70 ± 0.65 | 0.16 ± 0.07 | 0.71 ± 0.54 |
| WSOC | 1.3 ± 1.3 | 2.4 ± 1.5 | 0.73 ± 0.27 | 1.7 ± 1.3 | 0.70 ± 0.43 | 2.6 ± 1.4 |
| WSOC$_f$ | 0.26 ± 0.29 | 0.24 ± 0.32 | 0.18 ± 0.07 | 0.36 ± 0.37 | 0.09 ± 0.11 | 0.46 ± 0.32 |
| WSOC$_c$ | 1.0 ± 1.0 | 2.1 ± 1.2 | 0.55 ± 0.20 | 1.4 ± 1.0 | 0.60 ± 0.31 | 2.1 ± 1.1 |
| OC/EC | 4.0 ± 2.0 | 6.3 ± 3.7 | 3.2 ± 1.0 | 2.9 ± 1.1 | 3.2 ± 1.5 | 5.1 ± 3.5 |





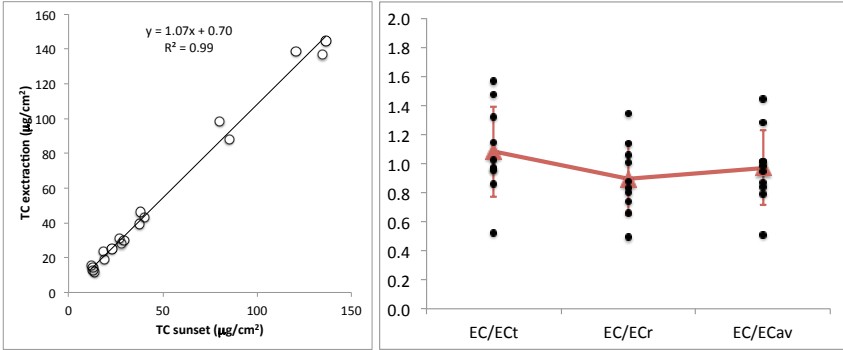

Figure 1: (a) Total carbon concentrations (TC) measured with a thermo-optical method (Sunset Analyzer) and the ACS system (extraction). (b) EC concentrations estimated from the ACS system using Eq. (3) and (4), divided by EC measured by a Sunset Analyzer using transmission correction ($M_{EC}/M_{ECt}$), reflection correction ($M_{EC}/M_{ECr}$) and by the average of $M_{ECr}$ and $M_{ECt}$ ($M_{EC}/M_{ECav}$). The red triangles represent averages over all data points and the error bars one standard deviation.





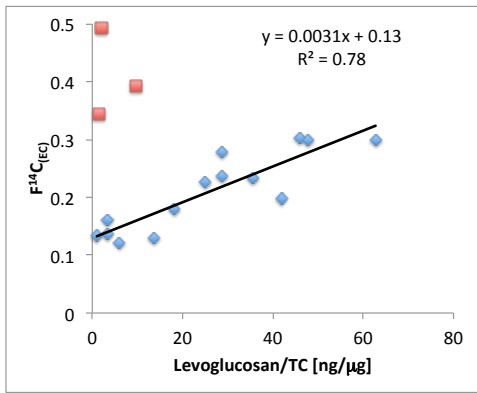

Figure 2: $F^{14}C_{(EC)}$ vs. levoglucosan mass fraction in the carbonaceous aerosol. The data marked in red are from three periods spring 2011, when high pollen concentrations were measured. The $F^{14}C$ data are not corrected for charring bias.



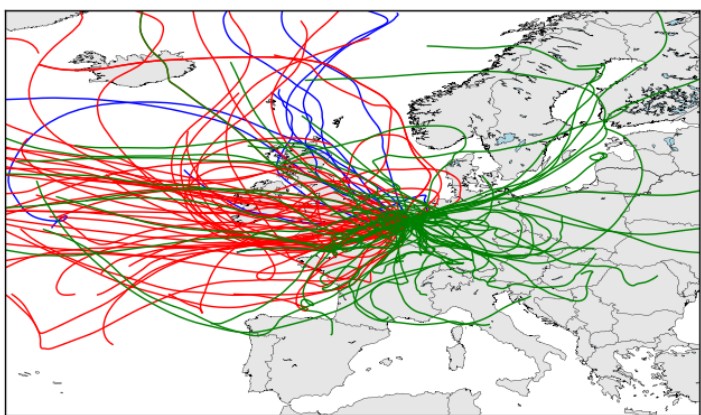

Figure 3: 48-hr air mass back trajectories (NOAA Hysplit) for the filter samples classified as modified

marine (red), marine (blue), and continental (green).



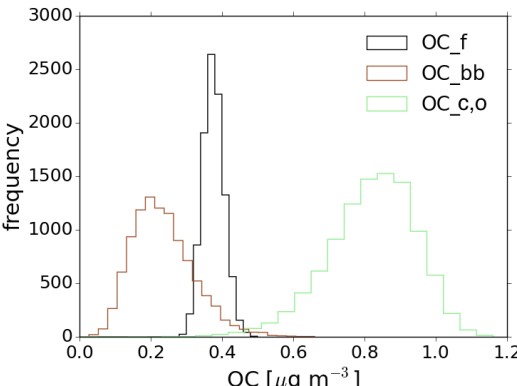

Figure 4: An example of the frequency distribution of OC concentrations from fossil-dominated, biomass burning, and biogenic sources for sample CA36, taken in June 2011. The concentrations of $OC_{bb}$ and $OC_{c,o}$ are not independent, larger concentrations of $OC_{bb}$ correspond to smaller concentrations in $OC_{c,o}$.





(a) (b)

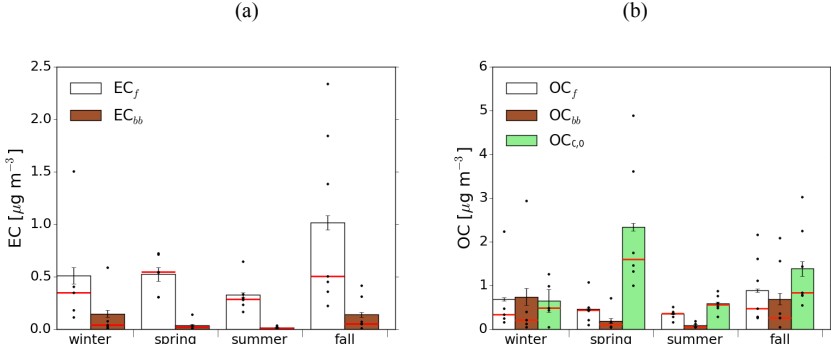

Figure 5. Concentrations of different sub-fractions of OC and EC from various sources averaged for winter, spring, summer, and fall in 2011/12. The bars show the average concentrations and the error bars the propagated uncertainties from the Monte Carlo simulations. The red line shows the median concentration. The individual data points are shown as black dots (a) EC from fossil-dominated sources ($EC_f$) and from biomass combustion ($EC_{bb}$), (b) OC from fossil-dominated sources ($OC_f$), biomass combustion ($OC_{bb}$), biogenic sources ($OC_{c,o}$).




(a)             (b)

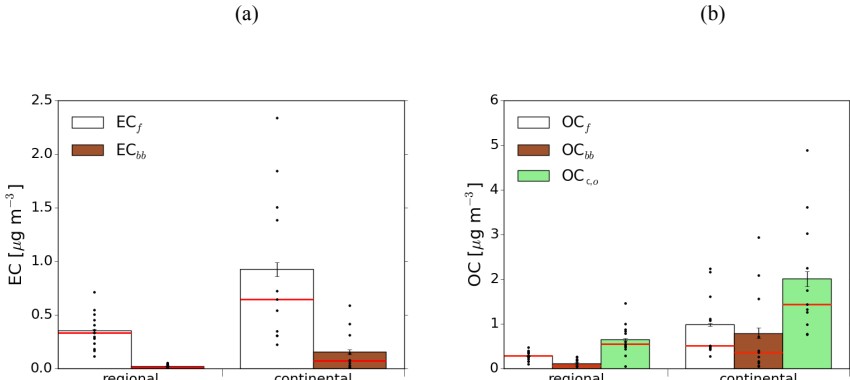

Figure 6. Concentrations of different sub-fractions of OC and EC from various sources in the time period from February 2011 to march 2012 averaged for regional and continental air mass origin. The bars show the average concentrations and the error bars the propagated uncertainties from the Monte Carlo simulations. The red line shows the median concentration. The individual data points are shown as black dots. (a) EC from fossil-dominated sources (EC$_f$) and from biomass combustion (EC$_{bb}$), (b) OC from fossil-dominated sources (OC$_f$), biomass combustion (OC$_{bb}$), and biogenic sources (OC$_{c,o}$).





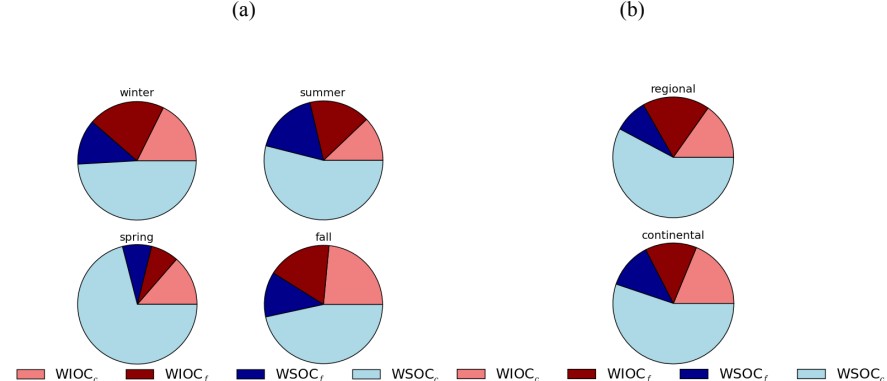

Figure 7. The contribution of water soluble and water insoluble organic carbon from fossil-dominated sources (WSOC$_f$ and WIOC$_f$) as well as water soluble and water insoluble organic carbon from contemporary sources (WSOC$_c$ and WIOC$_c$) to the total organic carbon. (a) Contribution of different carbon fractions during winter, summer, spring and fall. (b) Contribution of different carbon fractions for modified marine and continental air mass origin.



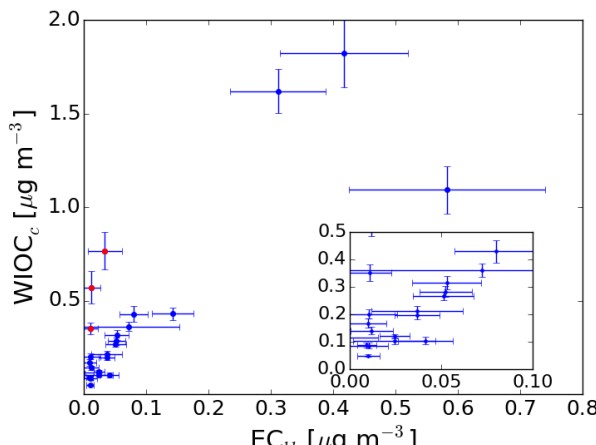

Figure 8. Scatter plot of contemporary water insoluble carbon concentrations (WIOC_c) against EC derived from biomass burning (EC_bb). EC_bb has been corrected for the contribution of highly refractory primary OC during three pollen events (red points).





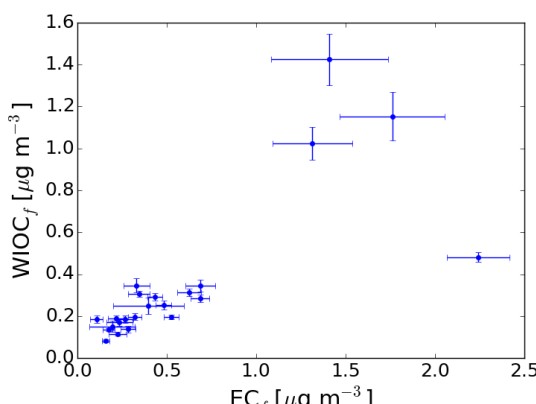

Figure 9. A scatter plot of EC concentrations from fossil-dominated sources (EC_f) versus
concentrations of water insoluble fossil carbon (WIOC_f).