# Peer review of "Sources and formation mechanisms of carbonaceous aerosol at a regional background site in the Netherlands: Insights from a year-long radiocarbon study"

_Atmospheric Chemistry and Physics, 2016_

## Referee Comment (RC1) · Anonymous Referee #1 · 27 Sep 2016

**General comments**

The authors present a comprehensive study regarding the sources of carbonaceous aerosol fractions in a regional background station in the Netherlands. Large focus is devoted to seasonal patterns in concentration and influence from the regional proximity in contrast to continental Europe.

I regard the scientific value of this work as moderate with the motivation that the used techniques and methods are not very novel. However, the authors prove the applicability of the method and techniques in a satisfactory manner. The presented results are clear and interesting. However, perhaps it is not that surprising that air masses from oceanic areas are cleaner than continental ones. In my understanding, there are limited of number of studies that have measured $F^{14}C$ in the presented carbon fractions, this study adds potentially significant information to this field.

The study is limited to the Netherlands and may be of national interest, however my impression is that there has been very few source apportionment studies of the carbonaceous aerosol conducted in this area. Hence, this study is of potential large interest for aerosol research in the Benelux area.

I lack an analysis regarding the "correctness" of the forecasted HYSPLIT trajectories. How did you assure that these forecasted trajectories were correct? Did you compare the forecasts to the actual trajectories (that "took place")?

I also lack a clear classification regarding the seasons. Did you classify them by calendar months, days, etc.? Or did you classify them by meteorological means, i.e. temperature? This is of crucial importance when interpreting the results.

Some effort can be spent on reviewing the acronyms used in this paper, there are many and some are very similar. A review and some corrections would increase the readability of the paper.

After some major corrections I believe that this manuscript should be published in ACP.

**Specific comments**

Page 2, line 12. Please replace "it" with what you actually mean, i.e. carbonaceous material.

Page 2, line 10-16. It would be nice if the authors could mention the fraction carbonaceous aerosol in PM10 or PM2.5 in Europe. To give the reader an idea of how large this fraction is.

Page 2, line 32. Can the authors please explain why the ratios are normalized to a $d^{13}C$ value of -25‰.

Page 3, line 13-19. You very nicely explain that the three major sources of carbonaceous aerosol are biogenic, fossil fuel and biomass burning derived. However, you only show references of biomass burning in the later section of the paragraph. I would like to see some references on studies that showed that fossil fuel aerosol mass is rather stable throughout the

year, further that the biogenic carbonaceous aerosol is totally dominating in rural areas during summer (Genberg et al. 2011; Yttri et al. 2011)

Page 3, line 35-37. This information should be given earlier in the introduction if this number is for Europe. If the number is relevant for the Netherlands, please ignore this comment.

Page 4, line 3-5. Perhaps omit this sentence. Also, I find this value rather low, I am not that surprised given the surrounding environment as you mention.

Page 4, line 15-21. Please state the altitude of the measurement station.

Page 4, line 15-21. Please state how you differentiated between the different seasons.

Page 4, line 23-28. Please state the flow of the high-volume sampler.

Page 4, line 26-28. How did you assure that the HYSPLIT forecasts were correct, and did you estimate the correctness of the forecasts? For me it is not unlikely that there were cases when the forecast said one thing, but the air masses did in fact arrive from another direction than was forecasted.

Page 5, line 4-30. Please clarify for all combustion steps the atmosphere used. Was it pure $O_2$ in all cases? Also, you did not measure carbon mass in these combustion steps, is that correct?

Page 5, line 18-20. How do you differentiate between the EC (in the OC-EC mixture) that is combusted in $450°C$ and EC combusted in $650°C$? Can you estimate the amount of EC evolved in the $450°C$ step?

Page 5, line 21-23. I don't understand how you derive the mean charring bias of 0.04? Please explain.

Page 5, line 31-36. Did you measure carbon mass on both of these facilities?

Page 6, line 7. Is it really 500 **mg**? That's a huge mass. Further, if you mean **μg**, I still question the number 500, perhaps you mean 50 μg?

Page 6, line 17. What do you mean by "Unknown samples"?

Page 6, line 23. Again, I question that you had 10-100 **mg**/cm$^2$ OC on your sample filters.

Page 7, line 7-15. In this paragraph I lack a motivation to why you should measure sugars in the first place. What types of sugars were your target compounds? I also lack some information that it is the levoglucosan that is of main interest here. Perhaps you can address this in the introduction or here in the method section? Also, what was the measurement uncertainty of the analysis?

Page 7, line 17-25. In this paragraph I lack information regarding the He-$O_2$ mixture, which proportions were used? Further, why did you use the QUARTZ protocol? What are the benefits by using this protocol instead of EUSAAR-2? What was the measurement uncertainty of the analysis?

Page 7, line 17-25. You should here state that you used TOA for comparison to ACS and perhaps the radiocarbon facilities to estimate carbon mass. After reading the whole method section I believed that you estimated the carbon mass by TOA, solely. However, when

reaching the result section, I found out that TOA was just a measure of comparison to ACS, is that correct? Either way, the carbon mass measurements needs to be clarified.

Page 7, line 28-29. This sentence should be presented earlier in the ACS method part.

Page 9, line 18. You have written $EC_{co}$, but do you mean $EC_f$?

Page 9, line 27. You have written $EC_{co}$, but do you mean $EC_f$?

Page 12, line 9. Levoglucosan should be mentioned earlier, in the introduction or in the proximity of the sugar measurements written in the method section.

Page 12, line 15-16. Here you mention glucose and sucrose. This should be mentioned earlier, in the introduction or in the proximity of the sugar measurements written in the method section.

Page 12, line 21-24. Please give a motivation why you chose to replace these values with values obtained from the regression line.

Page 12, line 30-32. This information should also be mentioned in the method section.

Page 12, line 40. In the figure caption of Figure 3 it says 48 h.

Page 13, line 10-14. In Figure 3, the blue lines were included into the red lines (modified marine)? Perhaps write this information in the figure caption.

Page 13, line 15-25. It is a bit confusing that you use "co" as an acronym for both "continental" and "contemporary, other". Consider changing this, it will most likely increase the readability of the paper.

Page 13, line 26-28. I think you should add the coverage in days to Table 2.

Page 14, line 32. Here you mention the seasonal pattern of $OC_{bb}$ concentration which is a bit confusing since this parameter is not presented in Table 3.

Page 15, line 23-24. Here it would be suitable with a reference.

Page 16, line 16-28. Again, here it would be nice to know how you classified the seasons. The difference between spring and fall should be small since you can expect these seasons be the intermediate of two extremes (i.e. winter and summer). However, this might not be the case depending on how you classified and defined your seasons. For increasing the interpretation and readability you should mention seasonal classification.

Page 16, line 39-40. How do you heat your residents in the Netherlands during winter? Is it non-aerosol producing energy source? Perhaps you can mention this somewhere.

Page 17, line 26. I assume you mean $\mu g$/m$^3$ and not **mg**/m$^3$?

Page 29, Table 1. It would be nice if you could add the references for these numbers in the table.

Page 31, Table 3. In the text you called "contemporary, other" "c,o", here you just call it "c". I would like to see consistency between the acronyms in the text and in the table.

Page 31, Table 3. Why did you merge $OC_{bb}$ and $OC_{c,o}$?

Page 32, Figure 1. I lack an explanation of the y-axis in Figure 1b.

Page 33, Figure 2. Is the equation valid only for the blue data points? Please clarify this in the figure caption.

**Technical corrections**

Page 12, line 20. Please replace "Weather" with "Whether".

Page 15, line 3. Please add "is". "Therefore we think it is unlikely……"

Whole document. Check for discrepancies between "c,o" and "c" acronyms. Including figures and tables.

**References**

Genberg et al. 2011. Source apportionment of carbonaceous aerosol in southern Sweden. ACP.

Yttri et al. 2011. Source apportionment of the summer time carbonacesous aerosol at Nordic background sites. ACP.

---

## Referee Comment (RC2) · Anonymous Referee #2 · 11 Oct 2016

The paper "Sources and formation mechanisms of carbonaceous aerosol at a regional background site in the Netherlands: Insights from a year-long radiocarbon study" by Dusek et al. (Manuscript number ACP_2016_624) is an interesting manuscript dealing with elemental and organic carbon fractions source apportionment in the Netherlands using 14C as a tracer for modern contributions. Radiocarbon measurements on OC and EC are still relatively scarce in the literature, especially covering a relatively long period such as the one covered here (1-year). The sampling strategy (1-week sampling) was well developed for the scopes of the manuscript which were mainly devoted to an overview of the sources of carbonaceous particles throughout the year, allowing

<**bold**>Printer-friendly version

<**bold**>Discussion paper

a good time coverage limiting the number of samples to be prepared and analysed.

In my opinion, the data presented in the paper merit publication on ACP, even if major revisions are needed in the text of the Results and Discussion sections. Indeed, in both of them the text lacks of numerical support to most of the adjectives/adverbs used in the sentences (e.g. high/low/relatively higher, slightly lower etc), making the text difficult to be read and understood. Numeric information justifying sentences can be however extracted in most cases by tables or figures, but complementary information should be added in the text to help the reader and to support statements (e.g. if absolute concentration are present in the table and in the text the sentence is "x is slightly higher than y" - what's slightly? - rephrasing as "x is slightly higher (zz%) than y" is an important help for paper clarity). Only in few cases, statements seem not to be supported by data. Such comments should be removed by the text.

In the following, detailed comments separated by pages are present.

Major revisions:

- page 5, line 22: "charred organic compounds contribute at most 5% to the recovered EC". Due to the thermal protocol chosen, it cannot be excluded a residual contribution also by resilient (not charred) organics.

- page 6: line 4: what do the authors mean with "virtually"? line 35: "The concentration and F14C of TC on the blank filter were calculated by adding the 35 carbon concentrations of OC and EC and were 0.68 $\mu$g cm-2 and 0.67, respectively": to retrieve F14C of TC, F14C of OC and EC should not be barely summed, but combined with OC, EC, and TC concentrations: TC F14C(TC) = OC F14C(OC)+ EC F14C(EC). But I assume from the numbers presented in lines 25-32 this is what has been done, just state better in the text.

-page 8: line 7-10: "(1) WIOC is completely recovered, which most likely results in an underestimate of WIOC, and (2) WIOC shows the same recovery as OC, which is

probably an overestimate, since WIOC is associated with more volatile primary organic material and usually less WIOC is lost to charring". I think this has to be phrased opposite: "(1) WIOC is completely recovered, which most likely results in an OVERESTIMATE of WIOC, and (2) WIOC shows the same recovery as OC, which is probably an UNDERESTIMATE, since WIOC is associated with more volatile primary organic material and usually less WIOC is lost to charring".

-page 9 line 31-32: "0.1 (modern carbon from biofuels contributes 10% to carbon emission from other combustion sources)" please rephrase: "0.1 (emissions of modern carbon from the biofuel added to road fossil fuels contributes 10% to carbon emission from such sources)"

- page 10 line 27: "the results are not strongly sensitive": what does "strongly sensitive" means? Please quantify at least with examples (i.e. varying in the min-max range the results modify of xxx%)

- page 11 line 21: "due to traces of water vapor or other impurities that are not removed entirely by the ACS method". To help the reader, it should be recalled the ACS quantifies TC manometrically.

- page 12 line 25: how was EC quantified in these samples? line 30: what does "relatively uncertain" means? In the end, was cooking accounted for in any way or not? If not, please rephrase "possible cooking contributions were not considered in the following" (if so, is there any literature study pointing at negligible contributions from cooking?)

- page 13 line 14: "and they were therefore included into the modified marine cases". Did the authors verify in any way that the results are not biased by this decision? lines 33-40: no data support the discussion. No a-priori assumption can be done on seasonal 14C concentration, as it is affected by two sources (biogenic and wood/biomass burning) which are predominant in different seasons, thus a priori considerations are not feasible. Moreover, temperature and total precipitation are not enough to determine

the extent of SOA formation (e.g. precursor concentrations and solar radiation intensity have a major role in SOA formation). Please remove all the discussion.

- page 14 line 1: "remained relatively constant for all seasons and air mass conditions, but was highest in spring". What does "relatively constant" stands for? Please rephrase: "were within xxx% (or within $\pm$yyy F14C) in all seasons, except in spring when they were yyy% higher" line 2: "F14C(EC) varied more strongly and was low in summer and high in winter". What does "more strongly", "high", and "low" stand for? Again, please give numerically indication. It is noteworthy that if compared e.g. to F14(OC), the terms "high" and "low" are nonsense unless further detail is given. line 15: "OCf was constrained in a relatively narrow range, whereas estimates for OCbb and OCc,o varied over a much wider range reflecting the large uncertainty in rbb". Please specify what "relatively narrow" and "much wider" stand for. line 25: "low": specify lines 27-29: "The main sources of fossil elemental carbon (ECf) in the Netherlands do not show a strong seasonal variation and its concentrations should therefore be relatively constant throughout the year". This is in contrast with what is said at line 26 ("higher planetary boundary layers in summer"), where a different dispersion condition depending on the season seems to be expected. Such seasonal variation would modify absolute concentrations in air of EC emitted by constant sources. lines 29-30: "However, there are relatively high ECf concentrations in fall": please, quantify "relatively high" line 30: "all other carbon fractions are elevated as well". Untrue (see table 3). Maybe the authors meant: "the fossil contribution of all the other carbon fractions is elevated as well (on average xxx%)". lines 31-36: please add numeric information throughout the text

page 15 lines 10-12: "the regional contribution in the Netherlands is relatively strong for OC and EC from traffic sources and the influence of long-range transport less important". What does "relatively strong" means? Please, quantify. line 14: "concentrations of ECbb are very low in regional pollution". How much lower compared to other conditions? lines 27-29: "The rainfall duration was on average 1 hr/day for continental conditions and 2 hrs/day for regional conditions and the amount was 1.2 vs 3.4 mm/day". Maybe it is more interesting the indication on the number of rainy days and the maximum rate in mm/h

Page 16 lines 31-32: "the concentrations of EC and OC are less variable and rather low". "Less variable": quantify (i.e. variability within xxx%). "Rather low": quantify absolute values and relative differences with other conditions. lines 33: "carbon concentrations within this low range also occur regularly under continental air mass conditions". What does "regularly" means? In how many cases compared to the total?

Page 17 line 9: "somewhat higher". Please quantify line 12: "WIOC consists to roughly equal parts of fossil and contemporary carbon with slightly higher fossil contributions in summer and slightly higher contemporary contributions in fall and winter". line 15-17: "The contributions of fossil and contemporary carbon fractions to OC (Figure 7b) do not change strongly for different air mass origins, even though the absolute concentrations of OC increased strongly in continental air masses". Please quantify "do not change strongly" (i.e. is within xxx%) and " increased strongly" (i.e. grew from xxx ug/m3 in regional air mass to yyy ug/m3 in continental air masses) line 26: "mg": sure? line 27: "most of the contemporary WIOC": lots of points in figure 8 have 0.1<WIOC<0.2 ug/m3. In such cases, contemporary WIOC from modern sources other than bb is far from being a small fraction of total contemporary WIOC.

Page 18 line 3-5: are the authors sure that no primary soluble organics are emitted by fossil fuel combustion? line 21: "relatively low": please, quantify line 23: " the variability of the WSOCf/ECf ratios is large": please, quantify

Page 19: line 5: " was highest in spring and lowest in summer": please, quantify

Page 20 line 15-19: "In contrast, WSOC is dominated by modern sources in all regions of the globe with usually only 0 –20% contributions from fossil sources (e.g., Kirillova et al., 2010, 2013, 2014; Szidat et al., 2006, 2009; Wozniak et al., 2012) This reflects that the main sources of modern OC, biomass burning and SOA formation, produce

largely water soluble carbon. The data from the Cesar site fall in this range with a fossil fraction of WSOC below 0.2 (Fig. 7)." The first sentence has no implication on the second one. Indeed WSOC being dominated by modern sources has no implication on WSOC/WIOC ratio of modern sources. Opposite, the second sentence is proven by contemporary WSOC domination in the total contemporary OC fraction.

Page 21, lines 29-32: "One of the most interesting results of our study is that, even though a large fraction of carbon emitted by biomass burning is water soluble, long-range transport of biomass smoke is the most important source of WIOCc in the Netherlands". Where is this point discussed? Just few words are mentioned in the text (page 15, line 12-13). "On the other hand, ECbb, WIOCc, and WSOCf increase by more than a factor of 4 under continental air mass conditions". When revising, please also consider the comment to page 17, line 27

Figure 3: more details on Hysplit use should be given (e.g. trajectory height, stability of the trajectories as function of starting point or beginning time)

Minor revisions:

- Page 6 line 7: "mg": Sure? line 17: "ultra-small samples larger than 2 $\mu$g C": larger or smaller? If "larger" is right, better to rephrase as "ultra-small samples down to 2 $\mu$g C" line 24: "mg": sure? line 30: change "from the five single filter pieces" in "from the five single blank filter pieces"

-Page 13 line 17: please change "last" in "previous" line 24: please remove "also"

---

## Author Comment (AC1) · 17 Jan 2017

The paper "Sources and formation mechanisms of carbonaceous aerosol at a regional background site in the Netherlands: Insights from a year-long radiocarbon study" by Dusek et al. (Manuscript number ACP_2016_624) is an interesting manuscript dealing with elemental and organic carbon fractions source apportionment in the Netherlands using 14C as a tracer for modern contributions. Radiocarbon measurements on OC and EC are still relatively scarce in the literature, especially covering a relatively long period such as the one covered here (1-year). The sampling strategy (1-week sampling) was well developed for the scopes of the manuscript which were mainly devoted to an overview of the sources of carbonaceous particles throughout the year, allowing a good time coverage limiting the number of samples to be prepared and analysed.
In my opinion, the data presented in the paper merit publication on ACP, even if major revisions are needed in the text of the Results and Discussion sections.
We thank the reviewer for the positive comments. We carefully considered all the comments of the reviewer in the revision of the manuscript.

Indeed, in both of them the text lacks of numerical support to most of the adjectives/adverbs used in the sentences (e.g. high/low/relatively higher, slightly lower etc), making the text difficult to be read and understood. Numeric information justifying sentences can be however extracted in most cases by tables or figures, but complementary information should be added in the text to help the reader and to support statements (e.g. if absolute concentration are present in the table and in the text the sentence is "x is slightly higher than y" - what's slightly? - rephrasing as "x is slightly higher (zz%) than y" is an important help for paper clarity). Only in few cases, statements seem not to be supported by data. Such comments should be removed by the text.
We thank the reviewer for the suggestions and the careful reading and commenting on the manuscript. After rereading the manuscript in the light of the reviewer's comments, we think the manuscript will be considerably improved by making the statements in the text more quantitative. In the original manuscript we tried to avoid citing numbers in the text that can easily be found in figures and tables, but in many occasions the numbers needed (e.g., in comparisons) are not directly available and we added them to the text as suggested.

For easier tracking of the changes, we marked all the changes in response to reviewer 1 in yellow and the changes in response to reviewer in grey throughout the revised manuscript.

In the following, detailed comments separated by pages are present.

Major revisions:
- page 5, line 22: "charred organic compounds contribute at most 5% to the recovered EC". Due to the thermal protocol chosen, it cannot be excluded a residual contribution also by resilient (not charred) organics.
This is true and later on in the manuscript we show that it indeed can occur. We added the following sentence to the manuscript: "Moreover, it is possible that highly refractory OC could survive the thermal treatment and combusted together with EC." (page 5, line 25-26)

- page 6: line 4: what do the authors mean with "virtually"?
> 95% (page 6, line 7)

line 35: "The concentration and F14C of TC on the blank filter were calculated by adding the 35 carbon concentrations of OC and EC and were 0.68 _g cm-2 and 0.67, respectively": to retrieve F14C of TC, F14C of OC and EC should not be barely summed, but combined with OC, EC, and TC concentrations: TC F14C(TC) = OC F14C(OC)+ EC F14C(EC). But I assume from the numbers presented in lines 25-32 this is what has been done, just state better in the text.

This is correct. We made the explanation more precise:
"The concentration of TC on the blank filters was calculated by adding the carbon concentrations of OC and EC and was 0.68 µg cm$^{-2}$. $F^{14}C_{(TC)}$ of the blank filters was calculated as the weighted average of $F^{14}C_{(OC)}$ and $F^{14}C_{(EC)}$ and was 0.67." (page: 6/7, lines: 37, 1-2)

-page 8: line 7-10: "(1) WIOC is completely recovered, which most likely results in an underestimate of WIOC, and (2) WIOC shows the same recovery as OC, which is probably an overestimate, since WIOC is associated with more volatile primary organic material and usually less WIOC is lost to charring". I think this has to be phrased opposite: "(1) WIOC is completely recovered, which most likely results in an OVERESTIMATE of WIOC, and (2) WIOC shows the same recovery as OC, which is probably an UNDERESTIMATE, since WIOC is associated with more volatile primary organic material and usually less WIOC is lost to charring".

The original phrasing is in principle correct, because if we assume that the recovery of WIOC is 100%, then (1) we assume that the measured WIOC equals the "real" total WIOC. If the recovery is less than 100%, then the "real" WIOC is calculated as measured WIOC/recovery, which is higher than (1). However, the way is formulated in the text is ambiguous and we see now that this is confusing. Therefore we changed the text to make this hopefully more clear:
"To estimate $M_{WIOC}$ we therefore assume two extreme cases: (1) WIOC is completely recovered: In this case $M_{WIOC}$ is equal to the mass of WIOC that was determined by the ACS system. This is most likely results in an underestimate of WIOC, because a fraction of WIOC might not combust at 360 °C. In this case the measured WIOC is less than the actual $M_{WIOC}$. (2) WIOC shows the same recovery as OC. In this case $M_{WIOC}$ is as the measured WIOC divided by the recovery of OC. This results probably in an overestimate of $M_{WIOC}$, since usually less WIOC than OC is lost to charring. We therefore consider a range of possible WIOC concentrations from a minimum of $M1_{WIOC}$ (complete recovery) to a maximum $M2_{WIOC}$ (recovery as OC). "(page: 8, lines 17-24)

-page 9 line 31-32: "0.1 (modern carbon from biofuels contributes 10% to carbon emission from other combustion sources)" please rephrase: "0.1 (emissions of modern carbon from the biofuel added to road fossil fuels contributes 10% to carbon emission from such sources)"
done (page 10, lines 15-16)

- page 10 line 27: "the results are not strongly sensitive": what does "strongly sensitive" means? Please quantify at least with examples (i.e. varying in the min-max range the results modify of xxx%)
We added: "If we shift the central value closer to $M1_{WiOC}$, e.g., $M1_{WIOC} + 1/3 * (M2_{WIOC} - M1_{WIOC})$, the average values of $WIOC_f$, $WIOC_c$ change by less than 5%." (page: 11 , lines8-9)

- page 11 line 21: "due to traces of water vapor or other impurities that are not removed entirely by the ACS method". To help the reader, it should be recalled the ACS quantifies TC manometrically.
We added: "and increase the sample pressure during the manometric determination of TC amount." (page: 12, lines 4-5)

- page 12 line 25: how was EC quantified in these samples?
Thank you for this comment. The EC was quantified as for the other samples in the original manuscript. We realize that this leads to a slight overestimation of EC and an underestimation of OC for these samples, because we should have subtracted the refractory biogenic OC from

the EC concentrations and add it to the OC. We have done this in the revised version of the manuscript. This leads to slightly adjusted numbers in table 3 for the spring and continental samples none of the concentration values change by more than 5%. Only the OC/EC ratio in the spring is now significantly higher. None of the conclusions or discussions of the manuscript are changed by this. (page: 13, line 3-4)

line 30: what does "relatively uncertain" means? In the end, was cooking accounted for in any way or not? If not, please rephrase "possible cooking contributions were not considered in the following" (if so, is there any literature study pointing at negligible contributions from cooking?)

We added the following clarification to the manuscript:
"although cooking usually emits much more OC than EC (e.g., Chow et al., 2004) and is probably not very relevant at a regional background site. Therefore any potential contribution from cooking is subsumed under ECbb in this study." (page: 13, lines 11-13)

- page 13 line 14: "and they were therefore included into the modified marine cases".
Did the authors verify in any way that the results are not biased by this decision?
Since it concerns only two data points it is obviously impossible to do a rigorous statistical test. Including these samples into the regional pollution case decreased the average TC concentration by only 0.1 $\mu g/m^3$. This makes sense considering that our sampling site is surrounded by major urban centers and highways and probably most of the aerosol we measure originates from these Dutch sources, both in the marine and modified marine case.

lines 33-40: no data support the discussion. No a-priori assumption can be done on seasonal 14C concentration, as it is affected by two sources (biogenic and wood/biomass burning) which are predominant in different seasons, thus a priori considerations are not feasible. Moreover, temperature and total precipitation are not enough to determine the extent of SOA formation (e.g. precursor concentrations and solar radiation intensity have a major role in SOA formation). Please remove all the discussion.
We decided to remove this discussion, following the suggestion of the reviewer.

- page 14 line 1: "remained relatively constant for all seasons and air mass conditions, but was highest in spring". What does "relatively constant" stands for? Please rephrase: "were within xxx% (or within _yyy F14C) in all seasons, except in spring when they were yyy% higher"
Thanks for this suggestion. We replaced the original statement with: "The average values of $F^{14}C_{(OC)}$ varied by ± 4% percent for winter, summer and fall as well as for continental and regional air mass conditions, but the average $F^{14}C_{(OC)}$ in spring (0.89) was 20% higher than the average of the rest of the seasons (0.74)." (page: 14, lines 15-17)

line 2: "F14C(EC) varied more strongly and was low in summer and high in winter". What does "more strongly", "high", and "low" stand for? Again, please give numerically indication. It is noteworthy that if compared e.g. to F14(OC), the terms "high" and "low" are nonsense unless further detail is given.
We changed this sentence to:
F14C increased by almost 70% from summer to winter (page: 14, line 17-18)

line15: "OCf was constrained in a relatively narrow range, whereas estimates for OCbb and OCc,o varied over a much wider range reflecting the large uncertainty in rbb".
Please specify what "relatively narrow" and "much wider" stand for.

Ok: We wrote instead: "$OC_f$ was constrained within 0.3 and 0.5 $\mu g/m^3$, whereas estimates for $OC_{bb}$ and $OC_{c,o}$ varied over a much wider range of roughly 0.5 $\mu g/m^3$ each, reflecting the large uncertainty in $r_{bb}$. (page 14, line 29-30)

line 25: "low":specify"
We added: "with average TC concentrations of 1.4 mg/m$^3$, which is less than half of the averages in other seasons" (page: 15; line: 1-2)

lines 27-29: "The main sources of fossil elemental carbon (ECf) in the Netherlands do not show a strong seasonal variation and its concentrations should therefore be relatively constant throughout the year". This is in contrast with what is said at line 26 ("higher planetary boundary layers in summer"), where a different dispersion condition depending on the season seems to be expected. Such seasonal variation would modify absolute concentrations in air of EC emitted by constant sources.

This sentence has been omitted in the revised version of the manuscript

lines 29-30:"However, there are relatively high ECf concentrations in fall": please, quantify "relatively high"

We now write: However, $EC_f$ concentrations in the fall are two times higher than in spring and winter and four times higher than in summer. (page: 15, line 6-7)

line 30: "all other carbon fractions are elevated as well". Untrue (see table 3). Maybe the authors meant: "the fossil contribution of all the other carbon fractions is elevated as well (on average xxx%)".

We changed this sentence to : "The concentrations $OC_f$ are elevated in fall as well" (page: 15, line 7 - 8)

lines 31-36: please add numeric information throughout the text

This paragraph reads now: The contemporary OC fractions are elevated in spring: $OC_c$ accounts for 70% and $WSOC_c$ for 60% of TC in spring, whereas $OC_c$ accounts only for roughly 50% and $WSOC_c$ for 35-40% of TC in other seasons. $EC_{bb}$ is more than 4 times higher in winter and fall than in spring and summer 2011. $WSOC_f$ is lower than $WSOC_c$ in all season except for summer. The standard deviations reflect the variability of pollution levels, which in winter are higher than the mean value. In summary, there are some indications of a seasonal variation in carbon concentrations, but the variability within each season is high and strongly influenced by weather and air mass conditions. (page: 15 , line 10-14)

page 15 lines 10-12: "the regional contribution in the Netherlands is relatively strong for OC and EC from traffic sources and the influence of long-range transport less important". What does "relatively strong" means? Please, quantify.

The regional contribution cannot be assessed quantitatively by this simple comparison, because it depends on too many other variables, however we can conclude that the regional contributions are higher for OC and EC from traffic sources than for other carbon fractions (but we cannot say how much higher exactly).
Therefore we changed this sentence to: "Our data therefore indicate that the regional contribution to OC and EC from traffic sources is higher than for other carbon fractions and the influence of long-range transport less important." (page: 15, line 29-31)

line 14: "concentrations of ECbb are very low in regional pollution". How much lower compared to other conditions?

This now reads: "Especially concentrations of $EC_{bb}$ are eight times lower in regional than in continental pollution" (page: 15, line 33)

lines 27-29: "The rainfall duration was on average 1 hr/day for continen- tal conditions and 2 hrs/day for regional conditions and the amount was 1.2 vs 3.4 mm/day". Maybe it is more interesting the indication on the number of rainy days and the maximum rate in mm/h

There were not so many sampling periods with completely dry conditions, but very often the rain was only of short duration. In this case the number of rainy days might be misleading, if a lot of them contained only a short period of precipitation, but would be counted similarly as days, where it rained most of the day.   In these conditions we feel that the total amount of precipitation that fell in each period (normalized by the number of days measured in each period) is a better measure for the influence of precipitation.

Page 16 lines 31-32: "the concentrations of EC and OC are less variable and rather low". "Less variable": quantify (i.e. variability within xxx%). "Rather low": quantify absolute values and relative differences with other conditions.

We changed this sentence to:
"the concentrations of EC and OC are below 0.5 mg m$^{-3}$ for EC and below 1 mg m$^{-3}$ for $OC_f$, $OC_{bb}$, $OC_{c,o}$ respectively, with average values less than half of the average concentrations encountered during continental air mass conditions. OC and EC concentrations are also less variable in recent pollution with relative standard deviations roughly 50% of the mean value of most carbon fractions, whereas standard deviations nearly approach the mean value for many carbon fractions continental air mass conditions." (page: 17, line 13-18)

lines 33: "carbon concentrations within this low range also occur regularly under continental air mass conditions". What does "regularly" means? In how many cases compared to the total?

We changed this sentence to:
"Carbon concentrations comparable to regional air mass conditions occur for $EC_f$, $EC_{bb}$ and $OC_{bb}$ in 50 – 60% of the continental samples and for $OC_f$ and $OC_{c,o}$ in 30 – 40% of the continental samples. This shows that despite higher average concentrations, continental air mass conditions do not always carry high concentrations of carbonaceous aerosol concentrations to the Netherlands." (page: 17, line 18-22)

Page 17 line 9: "somewhat higher".

We changed this to:
"The fraction of WSOC is the sum of the blue areas, which accounts for ~ 3/4 of the total OC in spring and summer and roughly 2/3 in fall and winter." (page: 17, line: 34-35)

Please quantify line 12: "WIOC consists to roughly equal parts of fossil and contemporary carbon with slightly higher fossil contributions in summer and slightly higher contemporary contributions in fall and winter".

Since the variability in the relative fossil contribution within each season is roughly 10% or more, we decided not to highlight the small variations and shortened the sentence to:
"WIOC consists to roughly equal parts of fossil and contemporary carbon." (page: 18, line 2).

line 15-17:
"The contributions of fossil and contemporary carbon fractions to OC (Figure 7b) do not change strongly for different air mass origins, even though the absolute concentrations of OC increased strongly in continental air masses". Please quantify "do not change strongly" (i.e. is within xxx%) and " increased strongly" (i.e. grew from xxx ug/m3 in

regional air mass to yyy ug/m3 in continental air masses)

We changed this sentence to:
"The contributions of fossil and contemporary carbon fractions to OC (Figure 7b) stay within 5% (absolute) for different air mass origins, even though the average concentrations of OC increased from 1 μg m-3 in regional to almost 4 μg m-3 in continental air masses." (page: 18, line 3-5)

line 26: "mg": sure?
corrected

line27: "most of the contemporary WIOC": lots of points in figure 8 have 0.1<WIOC<0.2 ug/m3. In such cases, contemporary WIOC from modern sources other than bb is far from being a small fraction of total contemporary WIOC.

This is a good point. We made the sentence more precise. It reads now:
"In other words, a large part of the variability in contemporary WIOC in the Netherlands seems to be associated with biomass combustion" (page: 18, line 14-15)

Page 18 line 3-5: are the authors sure that no primary soluble organics are emitted by fossil fuel combustion?

Usually, primary fossil OC is considered largely insoluble and our own data give direct evidence for this. $WSOC_f$ does not show any correlation with $EC_f$, especially if the highest concentration data points are omitted. These high concentration data points introduce a spurious correlation, because the concentrations of carbonaceous aerosol are generally very high for these four cases affecting most OC and EC fractions. This is now stated explicitly in text:
"In contrast, a similar linear regression of $WSOC_f$ against $EC_f$ yields an $R^2$ of 0.01, indicating that fossil water soluble WSOC does not have a common source with $EC_f$. If the four highest data points are included the $R^2$ is generally higher (0.92 for $WIOC_f$ and 0.48 for $WSOC_f$), but this is mainly due to the fact that for these four samples TC concentrations are much higher than average (ranging from 5 to 10 mg m$^{-3}$) and this leads to both higher OC and EC concentrations in general. It is not good practice to fit such bimodal data with a linear regression." (page: 18, line, 24-28)

line 21: "relatively low": please, quantify; line 23: " the variability of the WSOCf/ECf ratios is large": please, quantify

We made this paragraph more quantitative:
"The ratio of $WSOC_f/EC_f$ is 0.44 ± 0.46 under regional air mass conditions, which sample relatively fresh emissions. It increases to 0.6 ± 0.4 under continental air mass conditions, where older and more processed aerosol is sampled. The difference is significant at the 90% confidence level, but not at the 95% confidence level (p = 0.06). In general the large variability of the $WSOC_f/EC_f$ ratio indicates that $WSOC_f$ and $EC_f$ do not originate from a common source. (page: 19, line 11-16)

Page 19: line 5: " was highest in spring and lowest in summer": please, quantify

The values for spring (0.76) and summer (0.57) have been added to the text (page: 19, line: 33)

Page 20 line 15-19: "In contrast, WSOC is dominated by modern sources in all regions of the globe with usually only 0 –20% contributions from fossil sources (e.g., Kirillova et al., 2010, 2013, 2014; Szidat et al., 2006, 2009; Wozniak et al., 2012) This reflects that the main sources of modern OC, biomass burning and SOA formation, produce largely water soluble carbon. The data from the Cesar site fall in this range with a fossil fraction of WSOC below 0.2 (Fig. 7)." The first sentence has no implication on the second one. Indeed WSOC being dominated by modern sources has no implication on WSOC/WIOC ratio of modern sources. Opposite, the second sentence is proven by contemporary WSOC domination in the total contemporary OC fraction.

This is true. We deleted the sentence *"This reflects that the main sources of modern OC, biomass burning and SOA formation, produce largely water soluble carbon."* from the manuscript

Page 21, lines 29-32: "One of the most interesting results of our study is that, even though a large fraction of carbon emitted by biomass burning is water soluble, longrange transport of biomass smoke is the most important source of WIOCc in the Netherlands". Where is this point discussed? Just few words are mentioned in the text (page 15, line 12-13). "On the other hand, ECbb, WIOCc, and WSOCf increase by more than a factor of 4 under continental air mass conditions". When revising, please also consider the comment to page 17, line 27

This is a good point. We made this section of the text more specific:
"… long-range transport of biomass smoke acts to significantly increase the rather low background concentrations of $WIOC_c$ (around 0.1 mg m$^{-3}$) in the Netherlands. This can be concluded from the strong correlation of $WIOC_c$ with $EC_{bb}$ and that a strong increase in $EC_{bb}$ and $WIOC_c$ only happens during continental air mass conditions." (page: 22, line 21-24)

Together with the changed discussion with respect to the comment to page 17, line 27, this hopefully clarifies the point we wanted to make.

Figure 3: more details on Hysplit use should be given (e.g. trajectory height, stability of the trajectories as function of starting point or beginning time)

We added the trajectory end point height (50m) and a sentence concerning the temporal stability of the trajectories to the figure caption.

Minor revisions:
- Page 6 line 7: "mg": Sure?

corrected (page: 6, line 10)

line 17: "ultra-small samples larger than 2 _g C": larger or smaller? If "larger" is right, better to rephrase as "ultra-small samples down to 2 _g C"

done (page: 6, line 19-20)

line 24: "mg": sure?

corrected (page: 6, line 26)

line 30: change "from the five single filter pieces" in "from the five single blank filter pieces"

done (page: 6, line 33)

Page 13 line 17: please change "last" in "previous"

done (page: 13, line 39)

line 24: please remove "also"

done (page: 14, line 5)

---

## Author Comment (AC2) · 17 Jan 2017

General comments:

We thank the reviewer for the careful reading of the manuscript and the valuable remarks and comments. We have done our best to implement all the comments of the reviewer.

We changed the acronym for continental air mass (see below), and explain more clearly the distinction between OCc and OCc,o. However, if possible, we would like to keep the current acronyms. The acronym $F^{14}C$ was strongly recommended by Reimer et al., (2004) to denote fraction modern and we follow this convention. The most similar acronyms are OCc (all contemporary OC) and OCc,o (other contemporary OC, which denotes all contemporary OC, except OC from primary biomass burning; this is often denoted OCbio, but as one reviewer pointed out in the preliminary review, OCbio is misleading for this fraction). We find it difficult to come up with a very distinct, alternative acronym for this carbon fraction.

Please also not that in response to on comment by reviewer 2, we slightly changed the way quantify EC in the three spring samples presumably affected by pollen events. This changes some numbers in the tables and figures slightly, but does not change any of the discussion or conclusions.

For easier tracking of the changes, we marked all the changes in response to reviewer 1 in yellow and the changes in response to reviewer in grey throughout the revised manuscript.

RC: I lack an analysis regarding the "correctness" of the forecasted HYSPLIT trajectories. How did you assure that these forecasted trajectories were correct? Did you compare the forecasts to the actual trajectories (that "took place")?

**Answer:** We will make this more clear in the text: The sampling times were decided based on forecast trajectories, which are not always correct. We then used the actual back-trajectories (based on re-analysis data) to select some of the filters with the most consistent back trajectories for analysis. This is now explained in the paper (page: 13, lines: 22 - 24)

RC: I also lack a clear classification regarding the seasons. Did you classify them by calendar months, days, etc.? Or did you classify them by meteorological means, i.e. temperature? This is of crucial importance when interpreting the results.

**Answer:** We thank the reviewer for this useful comment. We classify the seasons by month: winter: Dec., Jan., Feb.; spring: March, April, May; summer: June, July, August; fall: Sept., Oct., Nov. This is now stated explicitly in the paper (page: 14, lines: 7-8).

Page 2, line 12. Please replace "it" with what you actually mean, i.e. carbonaceous material.
Done (page: 2, lines: 13)

Page 2, line 10-16. It would be nice if the authors could mention the fraction carbonaceous aerosol in PM10 or PM2.5 in Europe. To give the reader an idea of how large this fraction is.
We added the sentence: In Europe this fraction is typically between 30 and 60% of PM2.5

Page 2, line 32. Can the authors please explain why the ratios are normalized to a $d_{13}C$ value of -25‰.
This is to account for isotopic fractionation during sample pretreatment and measurement. This explanation is now added to the text (page: 2, lines: 33-34)

Page 3, line 13-19. You very nicely explain that the three major sources of carbonaceous aerosol are biogenic, fossil fuel and biomass burning derived. However, you only show references of biomass burning in the later section of the paragraph. I would like to see some references on studies that showed that fossil fuel aerosol mass is rather stable throughout the year, further that the biogenic carbonaceous aerosol is totally dominating in rural areas during summer (Genberg et al. 2011; Yttri et al. 2011)
We added the sentence about biogenic aerosol to the text. (page: 3, lines: 21-22). There is not a very clear reference that shows that fossil fuel aerosol is rather constant throughout the year, because not many long-tern studies exists.

Page 3, line 35-37. This information should be given earlier in the introduction if this number is for Europe. If the number is relevant for the Netherlands, please ignore this comment.
This number is relevant for the Netherlands and this is now clarified in the text (page: 4, lines: 1)

Page 4, line 3-5. Perhaps omit this sentence. Also, I find this value rather low, I am not that surprised given the surrounding environment as you mention.
This sentence is now omitted

Page 4, line 15-21. Please state the altitude of the measurement station.
0.7 m below sea level, this is now stated in the text (page: 4, lines: 17)

Page 4, line 15-21. Please state how you differentiated between the different seasons.
We classify the seasons by month: winter: Dec., Jan., Feb.; spring: March, April, May; summer: June, July, August; fall: Sept., Oct., Nov. This is note stated explicitly in the paper (page: 14, lines: 16-17).

Page 4, line 23-28. Please state the flow of the high-volume sampler.
500 l/min, this is now stated in the text (page: 4, lines: 26).

Page 4, line 26-28. How did you assure that the HYSPLIT forecasts were correct, and did you estimate the correctness of the forecasts? For me it is not unlikely that there were cases when the forecast said one thing, but the air masses did in fact arrive from another direction than was forecasted.
The sampling times were decided based on forecast trajectories, but after sampling we calculated actual back trajectories. Only a subset of samples, where the actual back trajectories were satisfactory was selected for analysis. Figure 3 shows the actual back-trajectories (based on re-analysis data) of the filters that were selected for analysis. This is now explained in the paper (page: 13, lines: 21-23)

Page 5, line 4-30. Please clarify for all combustion steps the atmosphere used. Was it pure $O_2$ in all cases? Also, you did not measure carbon mass in these combustion steps, is that correct?
On page 5, line 9 of the original manuscript it is stated that carbon fractions are

combusted in O2. In the revised manuscript we added this now for each carbon fraction. (page: 5 , lines: 15-20) The mass of carbon (OC; 360C step and EC; 650 °C) in each sample is determined manometrically. This is added to the text (page: 5, lines: 9-10)

 How do you differentiate between the EC (in the OC-EC mixture) that is combusted in 450◦C and EC combusted in 650◦C? Can you estimate the amount of EC evolved in the 450◦C step?

With the method as it is set up now, the CO2 evolved in the 450°C step is discarded and we do not measure the amount of carbon combusted in this step directly. However, because we also measure the amount and F14C of TC, we can estimate the amount of carbon evolved in this step as the difference between TC – OC(recovered at 360°C) – EC (recovered at 650°C). We can also make a crude estimate of what percentage of EC is recovered by comparing ECr to EC determined by the thermal optical method, or by the calculations outlined in section 2.8. The EC evolved at 450 is consequently ECtotal – ECr.

 I don't understand how you derive the mean charring bias of 0.04? Please explain.
This is a very rough calculation, assuming that 5% of recovered EC consists of charred OC and that $F^{14}C$ of OC is approximately 0.8. Since we cannot measure the charring directly for every sample, such a simple average correction for all the samples is the best we can do at the moment. We now add the explanation of how we arrive at 0.04 (page: 5, lines: 24-25)

 Did you measure carbon mass on both of these facilities?
The carbon mass was determined at the University of Utrecht on the ACS system, now explained in the manuscript (page: 5, lines: 9-10)

 Is it really 500 **mg**? That's a huge mass. Further, if you mean **μg**, I still question the number 500, perhaps you mean 50 μg?
Yes, thanks for spotting the error. Unfortunately all the micron symbols were lost, when we applied the ACP template to the manuscript, and even though we did our best to correct this, it seems we overlooked a few instances. Corrected (page: 6, lines: 10)

 What do you mean by "Unknown samples"?
"Unknown" was replaced by "The" (page: 6, lines: 20)

 Again, I question that you had 10-100 **mg**/cm2 OC on your sample filters.
Corrected (page: 6, lines: 26)

 In this paragraph I lack a motivation to why you should measure sugars in the first place. What types of sugars were your target compounds? I also lack some information that it is the levoglucosan that is of main interest here. Perhaps you can address this in the introduction or here in the method section? Also, what was the measurement uncertainty of the analysis?

We now give a brief motivation at the start of this paragraph: "In addition to $F^{14}C$ we also measured atmospherically relevant sugars, (e.g., levoglucosan, sucrose, glucose, mannosan). Levoglucosan can serve as an independent tracer of biomass burning and several other sugars can indicate primary bioglogical material, which cannot be traced by $^{14}C$ measurements alone." (page: 7, lines: 12-15)

The relative standard deviation of the measurements, determined based on replicate analysis of standards and blanks, was below 10%. (page 7; lines, 24-25)

Page 7, line 17-25. In this paragraph I lack information regarding the He-O$_2$ mixture, which proportions were used? Further, why did you use the QUARTZ protocol? What are the benefits by using this protocol instead of EUSAAR-2? What was the measurement uncertainty of the analysis?
The mixture contained 10% oxygen (page: 7,line: 31), the analytical uncertainty for OC and EC varied slightly with filter loading from 5% at loadings above 20 ug/cm2 to 7% for loadings around 10 ug/cm2 (page: 7, line: 34-36).
The Quartz Protocol is used routinely at the University of Vienna, where OC-EC measurements have been ongoing since before the EUSAAR protocol was introduced. Therefore the quartz protocol is still used to ensure comparability with previous measurements and older records. There was no specific scientific advantage for using it is this study.

Page 7, line 17-25. You should here state that you used TOA for comparison to ACS and perhaps the radiocarbon facilities to estimate carbon mass. After reading the whole method section I believed that you estimated the carbon mass by TOA, solely. However, when reaching the result section, I found out that TOA was just a measure of comparison to ACS, is that correct? Either way, the carbon mass measurements needs to be clarified.
We clarified this by changing the title of section 2.7. to "Estimation of OC and EC by thermal-optical analysis" and of section 2.8 to "Estimation of OC, EC, WIOC concentrations based on carbon fractions recovered by the ACS system" to make clear that we estimate OC and EC in two different ways.
To section 2.7. we added the sentence "EC, OC and TC measured by the thermal optical method are used to evaluate the estimated values of OC and EC based on the data from the ACS system (see section 2.8)." (page: 8, lines 1-3)

Page 7, line 28-29. This sentence should be presented earlier in the ACS method part.
In section 2.3 (page 5, line 18 of the original manuscript) we write: "Then the oven temperature is raised to 450 °C for two minutes and in this step a mixture of the most refractory OC and EC is burned off the filter." We opt keep this sentence and remove the corresponding sentence in section 2.8 in order not to mention this twice.

Page 9, line 18. You have written EC$_{co}$, but do you mean EC$_f$?
Yes, thank you for spotting this typo, we have corrected this (page: 10, line: 2)

Page 9, line 27. You have written EC$_{co}$, but do you mean EC$_f$?
Also corrected (page: 10, line: 11)

Page 12, line 9. Levoglucosan should be mentioned earlier, in the introduction or in the proximity of the sugar measurements written in the method section.
done (page: 7 , lines 12-14)

Page 12, line 15-16. Here you mention glucose and sucrose. This should be mentioned earlier, in the introduction or in the proximity of the sugar measurements written in the method section.
done (page: 7 , lines 12 - 15)

Page 12, line 21-24. Please give a motivation why you chose to replace these values with values obtained from the regression line.

We added the following explanation: This can be seen as a crude correction for the highly refractory part of OC that was apparently incorrectly classified as EC from biomass burning. Without this correction we would overestimate the contribution of biomass burning to the carbonaceous aerosol in spring. (page: 13, lines 4-6)

Page 12, line 30-32. This information should also be mentioned in the method section.
This information is present in section2.3 (page: 5, lines 20-23 in the original manuscript). Therefore we changed the discussion in section 3.1, by referring to section 2.3, and just pointing out that the data shown in Figure 2 are not yet bias corrected. (page: 12, lines 29-30)

Page 12, line 40. In the figure caption of Figure 3 it says 48 h.
Thank you for spotting this mistake, it should say 96 hours in the figure caption.

Page 13, line 10-14. In Figure 3, the blue lines were included into the red lines (modified marine)? Perhaps write this information in the figure caption.
Done

Page 13, line 15-25. It is a bit confusing that you use "co" as an acronym for both "continental" and "contemporary, other". Consider changing this, it will most likely increase the readability of the paper.
Yes, we changed co to cont for continental conditions in the tables.

Page 13, line 26-28. I think you should add the coverage in days to Table 2.
This is a good suggestion, we added this information.

Page 14, line 32. Here you mention the seasonal pattern of $OC_{bb}$ concentration which is a bit confusing since this parameter is not presented in Table 3.
Thank you for pointing this out. We deleted OCbb here.
Page 15, line 23-24. Here it would be suitable with a reference.
We added a reference (Ohata et al., 2016) (page: 16, lines 4)

Page 16, line 16-28. Again, here it would be nice to know how you classified the seasons. The difference between spring and fall should be small since you can expect these seasons be the intermediate of two extremes (i.e. winter and summer). However, this might not be the case depending on how you classified and defined your seasons. For increasing the interpretation and readability you should mention seasonal classification.
We added this information (page: 14 , lines 7-8)

Page 16, line 39-40. How do you heat your residents in the Netherlands during winter? Is it non-aerosol producing energy source? Perhaps you can mention this somewhere.
Heating in the Netherlands is done to a large part with gas, which produces very little aerosol. We added a brief discussion of the main sources of ECf in section 3.2 (page: 15, lines 3-5)

Page 17, line 26. I assume you mean $\mu g/m_3$ and not $mg/m_3$?
Thank you. corrected (page: 18, lines 13)

Page 29, Table 1. It would be nice if you could add the references for these numbers in the table.
Here we would prefer to keep the references in text, because not all the references are treated equally, e.g. we cite Szidat and references therein as a basis of our estimate and

adjust the values in this publication, considering several newer publications. Therefore, it would be somewhat misleading to just give the references in the table without all the explanations added in the text.

Page 31, Table 3. In the text you called "contemporary, other" "c,o", here you just call it "c". I would like to see consistency between the acronyms in the text and in the table.
As defined in section 2.9 (page 10, line 32 ff), the symbol c stands for all contemporary carbon, i.e., the sum of biomass burning and contemporary other carbon. $OC_c = OC_{bb} + OC_{c,o}$. We will add this to the figure legend to avoid misunderstandings.

Page 31, Table 3. Why did you merge $OC_{bb}$ and $OC_{c,o}$?
Because as can be seen in figure 3, the separation between $OC_{bb}$ and $OC_{c,o}$ is actually more semi-quantitative, and in the table we prefer to give concentrations of carbon fractions that we can determine with greater certainty. This is now mentioned in the text. (page: 14 , lines 38-39)

Page 32, Figure 1. I lack an explanation of the y-axis in Figure 1b.
We added a better explanation in the caption of figure 1. We also added the y-axis lable "ratio" to indicate that the ratios of ACS EC and Sunset EC are plotted.

Page 33, Figure 2. Is the equation valid only for the blue data points? Please clarify this in the figure caption.
Done

**Technical corrections**
**all the technical corrections have been corrected**
Page 12, line 20. Please replace "Weather" with "Whether".
done (page: 12, line: 39)

Page 15, line 3. Please add "is". "Therefore we think it is unlikely……"

done (page: 16, line: 23)

Whole document. Check for discrepancies between "c,o" and "c" acronyms. Including figures and tables.
We checked very carefully and could not find any discrepancies (to our knowledge). Everywhere c refers to all contemporary carbon (i.e. sum of biomass burning and biogenic carbon) and c,o to "other contemporary carbon (i.e. all contemporary OC, except primary $OC_{bb}$).

**References**
Genberg et al. 2011. Source apportionment of carbonaceous aerosol in southern Sweden. ACP.
Yttri et al. 2011. Source apportionment of the summer time carbonace